# Glacier retreat creating new Pacific salmon habitat in western North America

Kara J. Pitman [1✉], Jonathan W. Moore[1], Matthias Huss [2,3,4], Matthew R. Sloat [5], Diane C. Whited[6], Tim J. Beechie[7], Rich Brenner[8], Eran W. Hood[9], Alexander M. Milner [10,11], George R. Pess[12], Gordan H. Reeves[13] & Daniel E. Schindler[14]

Glacier retreat poses risks and benefits for species of cultural and economic importance. One example is Pacific salmon (*Oncorhynchus* spp.), supporting subsistence harvests, and commercial and recreational fisheries worth billions of dollars annually. Although decreases in summer streamflow and warming freshwater is reducing salmon habitat quality in parts of their range, glacier retreat is creating new streams and lakes that salmon can colonize. However, potential gains in future salmon habitat associated with glacier loss have yet to be quantified across the range of Pacific salmon. Here we project future gains in Pacific salmon freshwater habitat by linking a model of glacier mass change for 315 glaciers, forced by five different Global Climate Models, with a simple model of salmon stream habitat potential throughout the Pacific Mountain ranges of western North America. We project that by the year 2100 glacier retreat will create 6,146 (±1,619) km of new streams accessible for colonization by Pacific salmon, of which 1,930 (±569) km have the potential to be used for spawning and juvenile rearing, representing 0 to 27% gains within the 18 sub-regions we studied. These findings can inform proactive management and conservation of Pacific salmon in this era of rapid climate change.

[1] Earth to Ocean Research Group, Simon Fraser University, Burnaby, BC, Canada. [2] Laboratory of Hydraulics, Hydrology and Glaciology (VAW), ETH Zurich, Zurich, Switzerland. [3] Department of Geosciences, University of Fribourg, Fribourg, Switzerland. [4] Swiss Federal Institute for Forest, Snow and Landscape Research (WSL), Birmensdorf, Switzerland. [5] Wild Salmon Center, Portland, OR, USA. [6] Flathead Lake Biological Station, University of Montana, Polson, USA. [7] Watershed Program, Fish Ecology Division, Northwest Fisheries Science Center, NOAA Fisheries, Seattle, WA, USA. [8] Alaska Department of Fish and Game, Division of Commercial Fisheries, Juneau, AK, USA. [9] Program on the Environment, University of Alaska Southeast, Juneau, AK, USA. [10] School of Geography, Earth and Environmental Science, University of Birmingham, Edgbaston, Birmingham, UK. [11] Institute of Arctic Biology, University of Alaska, Fairbanks, AK 99775, USA. [12] Fish Ecology Division, Northwest Fisheries Science Center, National Marine Fisheries Service, NOAA Fisheries, Seattle, WA, USA. [13] USDA Forest Service, Pacific Northwest Research Station, Corvallis, OR, USA. [14] School of Aquatic and Fishery Sciences, University of Washington, Seattle, WA, USA. ✉email: karapitman@gmail.com

Climate change is driving rapid changes in Earth's ecosystems with new challenges and opportunities in resource management. For example, the loss of Arctic ice is posing risks to culturally important species like polar bears[1], but also creating frontiers for emerging fisheries[2]. One group of species being strongly impacted by climate change is migratory Pacific salmon (*Oncorhynchus* spp)[3]. Although Pacific salmon abundances have shifted from region to region over decades to centuries in response to climatic variability[4], ocean heat waves, low summer water flows, and excessively warm water temperatures are currently stressing many wild salmon populations[3,5]. At the same time, the warming of Arctic and subarctic freshwaters[6] and contemporary glacier retreat[7,8] are creating potential new frontiers for salmon. While glacier retreat can have a variety of direct and indirect impacts on salmon ecosystems[9–12], over the next century retreat of glacier ice will create new streams that, if not too steep for salmon migration, can provide future salmon habitat. For example, pink salmon abundance grew to >5000 adult spawners within ~15 years of a new stream (~2 km) and lake system being created following glacier retreat in Glacier Bay, Alaska[8].

Although salmon colonization of recently deglaciated streams has been well documented in individual watersheds[8], predicting future shifts in the distribution of productive salmon habitat remains a challenge, and there are no regional projections for the creation of new salmon habitat in response to retreating glaciers. Forecasting the location of emerging salmon habitat is imperative because, while declining glacier ice can present local opportunities for salmon, it is also creating new prospects for large-scale resource extraction industries such as mining, which have the potential to degrade these salmon habitat frontiers[13–16]. Understanding the timing and location of emerging salmon habitat frontiers throughout the Pacific Mountain ranges of western North America can inform forward-looking management decision-making and conservation planning.

The ~46,000 glaciers in the Pacific mountain ranges of North America cover an area of ~81,000 km[2,17], of which 80% fall within the range of Pacific salmon (Fig. 1a). These glaciers are rapidly declining in volume, thickness, and area, accelerated by recent anthropogenic climate warming[18–20]. For example, between 2006 and 2016, glaciers in western Canada lost an average of 1% of their ice mass annually[21] and are projected to lose up to 80% of their ice volume by 2100 in some regions[22].

Here we quantify emerging salmon streams created from glacier retreat, by using Digital Elevation Models within a Geographic Information Systems framework to derive a synthetic stream network for glacierized watersheds in the 623,000 km$^2$ region extending from southern British Columbia to southcentral Alaska (Fig. 1a). Synthetic stream networks include both present-day and future salmon streams (Methods). Using stream gradient-based salmon migration thresholds, we identify which glaciers are accessible to salmon (Methods). For the accessible glaciers within the 18 sub-regions of our study region, we model the timing of glacier retreat[23] and derive future stream networks based on sub-glacial terrain (Fig. 1b). Modeled glacier retreat was driven by temperature and precipitation projections from an ensemble of five Global Climate Models (GCM; Methods) forced by two climate emission scenarios (Representative Concentration Pathways, RCP), RCP4.5 and 8.5, under which global emissions are expected to peak at ~2050 and after 2100, respectively[24]. We present both scenarios but focus on the more moderate RCP4.5.

Here, we show where and when glacier retreat will create thousands of kilometers of new streams accessible for colonization by Pacific salmon, many of which are potentially suitable for spawning and juvenile rearing. We quantify two dimensions of future salmon habitat: the total future salmon-accessible stream kilometers (kms) and, of this amount, the habitat suitable for spawning and juvenile rearing. First, we quantify the extent of the stream network colonizable by adult migrating salmon based on two stream gradient thresholds inhibiting adult salmon

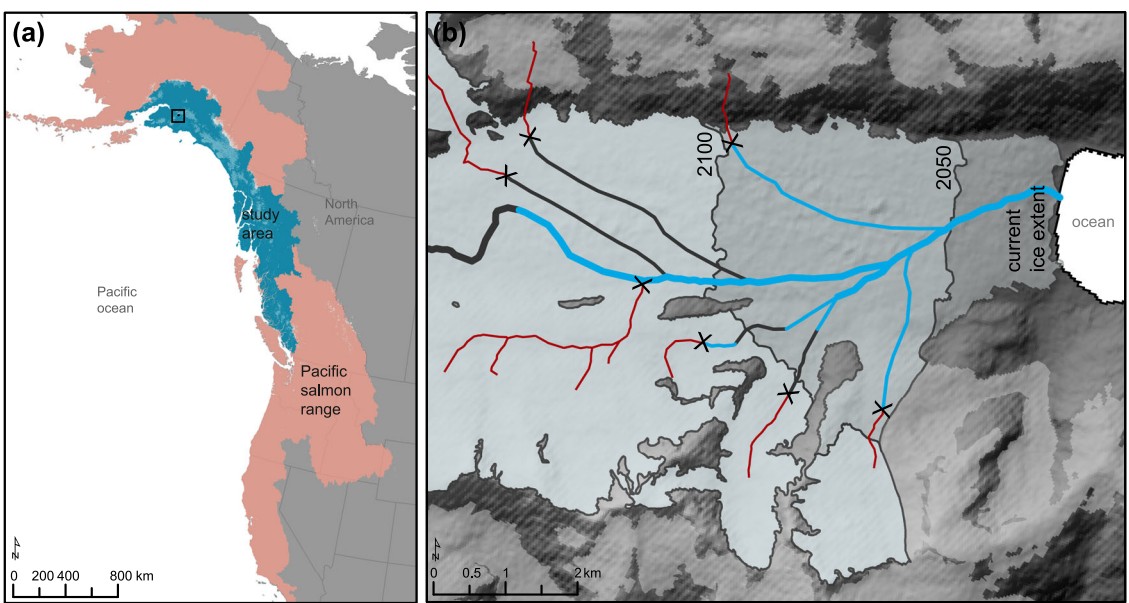

**Fig. 1 Pacific salmon range in glacierized watersheds and how glaciers will create new streams. a** Map showing the Pacific salmon range in North America (pink), and our study region (blue). Glacier outlines are in grey. The black box indicates the location of example focal area, shown in **b** Harriman Glacier, Prince William Sound, Alaska, showing approximate glacier retreat (for the benchmark years 2050 and 2100), future salmon-accessible streams (<10% stream gradient threshold over ~500 m; blue and black), and suitable habitat below a 0–2% stream gradient over ~500 m (blue). Thicker lines represent higher stream orders and narrower lines represent lower stream orders. Streams >10% stream gradient threshold are marked with an X and colored in red.

migration. These gradient thresholds are guided by an analysis of salmon presence records and stream network geomorphology in the Susitna River, a large watershed (53,000 km$^2$) within our study region with extensive data on the spatial extent of different salmon species (Supplementary Fig. 1; Supplementary Table 1; Methods). In the Susitna River watershed, the occurrence of most salmon species is common in accessible stream reaches up to a gradient threshold of 10% and the occurrence of all species is rare in reaches above a gradient of 15% (Supplementary Fig. 1). This range of gradient thresholds is supported by past studies (Methods)[25–27]. Thus, we select the conservative (10%) and a more inclusive (15%) stream gradient threshold for our salmon accessibility analysis. Second, we estimate the extent of new salmon-accessible streams that are second-order and greater (Methods), with lower gradients ranging from 0 to 2% and 0–4%, bracketing less and more inclusive bounds of preferred salmon spawning and juvenile rearing habitat, respectively (Methods; Supplementary Table 2)[26,28–30]. Thus, this second step defines the extent of newly accessible streams within gradient ranges associated with productive salmon habitat. While a variety of factors, such as changing hydrology, water temperature, and sediment dynamics will continue to change following glacier retreat and ultimately determine habitat suitability for salmon[9–12], here we quantify the lengths of new salmon-accessible streams and potential salmon habitat that will be available following glacier retreat by 2050, 2100, and under potential complete deglaciation.

## Results and discussion

**Glacier retreat creating future salmon-accessible streams**. We identified 315 retreating glaciers at the headwaters of present-day streams that will create salmon-accessible streams assuming a 10% stream gradient threshold for upstream salmon migration, and 603 glaciers assuming a 15% stream gradient threshold (Supplementary Table 1). Although the number of glaciers currently covering salmon-accessible streams is proportionally low, given that there are ~46,000 glaciers in the study region, salmon-accessible glaciers are particularly large, representing ~50% of the total glacier area in the study region regardless of which stream gradient threshold is used to define salmon accessibility. The total number of salmon-accessible glaciers roughly doubles with a 15% stream gradient threshold; however, these additional glaciers are small and represent a minimal increase in total glacier area (Supplementary Table 1).

Over the entire study region, we estimate an increase of between 6146 (±1619; this and the following uncertainty corresponds to ± one standard deviation and originate from: GCM projections, ice thickness estimates, and stream segment length) kms and 9296 (±2740) kms of future salmon-accessible streams by 2100 using the RCP4.5 climate scenario under the 10% (Fig. 2) and 15% (Supplementary Fig. 2; Supplementary Table 3) gradient thresholds, respectively. The projected increase in salmon-accessible streams was not evenly distributed across the 18 sub-regions of our study region, both in terms of absolute and proportional habitat gains. For example, our analysis indicates that seven out of 18 sub-regions show negligible to no gains in salmon habitat because most contemporary glaciers in these sub-regions have already retreated above the limits of upstream salmon migration (Fig. 2). In contrast, we project that the Gulf of Alaska sub-region will have an additional 2622 (±764) kms (27% increase) of salmon-accessible streams under the conservative stream gradient threshold of 10% (Fig. 2; Supplementary Table 3). In some sub-regions, we project substantial absolute gains in salmon-accessible stream kms even though the proportional increase is relatively small. For example, the Copper River will gain a projected 1064 (±344) salmon-accessible stream kms, but

this represents only a 2% increase within this large watershed. In general, our analysis indicates that the greatest gains in salmon-accessible streams will occur in areas where large glaciers occupy low gradient terrain near the coast.

The timing of when future salmon-accessible streams are exposed depends on the modeled rate of glacier retreat across the 18 sub-regions investigated (Fig. 2). Of the total salmon-accessible stream kms that could be gained with potential complete deglaciation, 23% will be gained by 2050, and 63% by 2100. There were pronounced regional differences, for example, the limited salmon-accessible stream gains projected for the Skeena River watershed and North Coast of BC will all be created by 2050. In contrast, in the Gulf of Alaska sub-region, which contains some of the largest remaining icefields in North America, only ~20% of new stream kms will be created by 2050. While we propagated the uncertainty within GCM projections, ice thickness estimates, and stream segment length, there are additional uncertainties within the glacier retreat model (GloGEM) used in our analysis. An intercomparison of global glacier models indicated that the model used in this study yields somewhat faster area loss than other models in the second half of the 21st century although projected ice volume is close to the median[19]. Further, the different climate scenarios affect the projected rates of future stream creation, with the RCP8.5 scenario projecting faster rates of glacier retreat in 2100 and thus earlier potential salmon-accessible stream gains in all 18 sub-regions (Supplementary Table 3). Thus, the unknown future pace of climate change as controlled by greenhouse gas emissions and feedbacks in the Earth-Atmosphere climate system will impact the rate of this habitat creation.

**Emerging habitat for Pacific salmon habitat for spawning and juvenile rearing**. A subset of the created future salmon-accessible streams will possess geomorphic conditions associated with favorable salmon spawning and juvenile rearing habitat. We conservatively estimate that glacier retreat will create 1930 (±569) kms of future salmon spawning and rearing habitat (gradient threshold <10%, 0–2% gradient for spawning and rearing habitat) by 2100 under RCP4.5 (Fig. 3) throughout the entire study region. The Gulf of Alaska and Copper River sub-regions have the largest projected increase in salmon spawning and rearing habitat, with 757 (±279) and 408 (±105) kms, respectively, by 2100 (Fig. 3). Watershed topography exerts a strong control on future habitat expansion. For example, projections over the entire study region with steeper stream gradient thresholds for future rivers (<15%) and habitat (0–4%) show 65% more habitat gained than the conservative estimates, with some sub-regions having more gains than others (Fig. 3, Supplementary Table 4). Thus, future increases in salmon habitat will vary depending on the specific topography of each watershed and the swimming abilities of each salmon species to access these new habitats.

Gains in salmon habitat are substantial enough that they could lead to new sizable increases in salmon production in some locations. For example, one km of suitable stream habitat can produce ~500–1500 juvenile coho salmon (Supplementary Table 5)[31]. Thus, with hundreds to thousands of kms of new habitat being created from glacier retreat, there is a potential to produce hundreds of thousands to millions of additional juvenile salmon, depending on species[32]. Despite potentially interacting freshwater conditions that can influence salmon productivity such as water temperature and flow, sediment supply and stability, and stream morphology, juvenile salmon production is strongly and positively related to available stream length across diverse watersheds[31]. Thus, as the extent of stream habitat increases, the number of salmon in those streams should

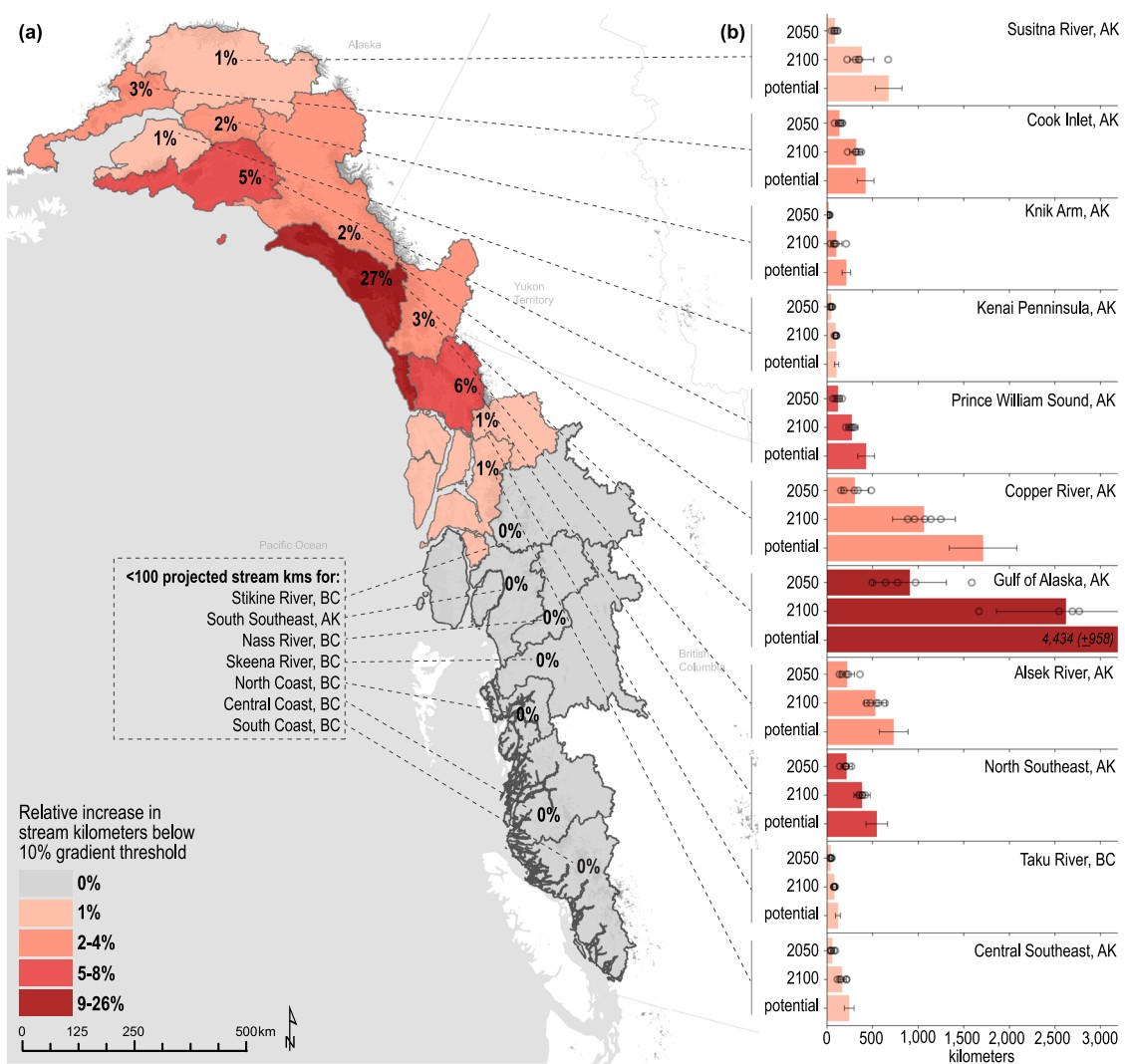

**Fig. 2 Projected future salmon-accessible stream kms with <10% stream gradient migration threshold. a** Map showing projected percent increase in future salmon-accessible stream kms, below a 10% stream gradient threshold for 2100, relative to present-day stream kms summed for each of the 18 sub-regions. Colors are defined in map. Glacier retreat projections, in response to five GCMs with RCP4.5 emission scenario, are used as an ensemble mean. **b** Bar plots representing ensemble-mean projected salmon-accessible stream kms with <10% stream gradient threshold for the years 2050, 2100, and potential complete deglaciation (i.e., once glaciers have retreated completely from the landscape) for each of the 18 sub-regions having >0% increase in future salmon-accessible stream kms. Projections are computed from 10-year averages centred ~2050 and 2100. Colors reflect percent increase values presented in A. Error bars correspond to ensemble-mean ± one standard deviation and originate from: GCM projections (RCP4.5, for 2050 and 2100), ice thickness estimates, and stream segment length (see Methods section "Uncertainty estimates'). Points represent projections for the individual GCMs for 2050 and 2100 (N = 5). Projected future salmon-accessible stream kms (±one standard deviation) are presented in the Source Data File.

generally increase. Unless overshadowed by larger-scale declines in salmon productivity, such as due to local decreases in water quality and supply or broad-scale decreased ocean survival[33,34], these forecasted increases in stream kms could lead to emerging salmon fisheries of local importance when adults return to spawn. For example, over the last century, glacier retreat in the Kenai Peninsula, Alaska, led to the establishment of sockeye salmon populations that support a local commercial fishery[35]. While salmon populations are controlled by many factors across their life cycles that are shifting with climate change[3], areas with increases in the extent of freshwater habitat represent new hotspots of potential increased salmon production.

**Downstream effects of glacier retreat and broader context.** Approximately half of the glacier area within our study region occurs in steep, mountainous terrain that is inaccessible to

migratory salmon, particularly in British Columbia, Canada (Figs. 2 and 3, Supplementary Table 1). However, the decrease of runoff from these perched alpine glaciers will impact the quality of downstream salmon habitat[10,12,36]. Across the range of Pacific salmon, southern arid regions will be challenged as glacier meltwater diminishes. This meltwater can sustain water flow and provide cool water that supports salmon across their life cycle, particularly during the summer months, and its loss could decrease the quality and quantity of salmon habitat[12,37,38]. In contrast, southcentral Alaska regions, such as the Copper River, will experience substantial increases in glacier meltwater during the summer months over the coming years due to the extensive network of glaciers that contribute flow to pro-glacial rivers[39]. Increases in glacier meltwater could lead to more cold water and higher turbidity levels in downstream rivers in the next few decades, which could either challenge or improve the habitat for salmon[12]. Thus, the retreat of inaccessible glaciers will have

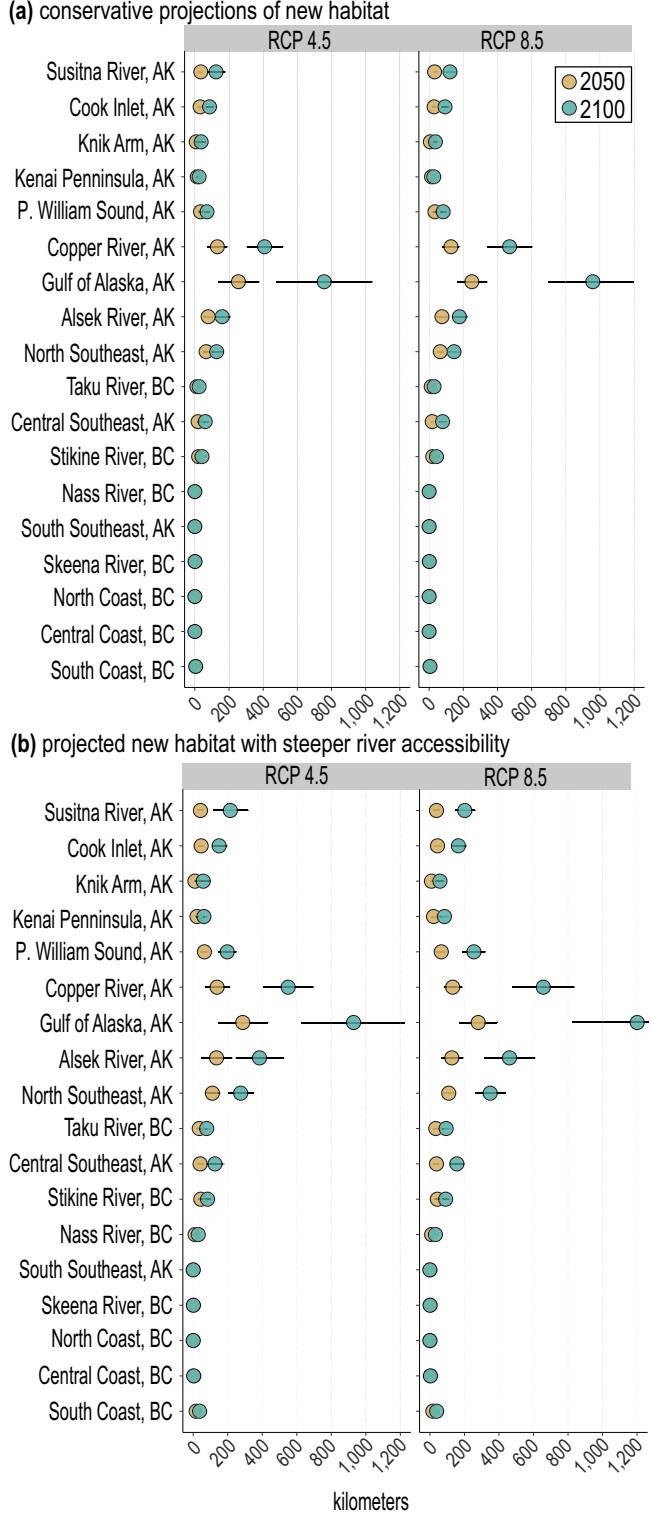

**Fig. 3 Projected kms suitable for salmon spawning and juvenile rearing habitat by 2050 and 2100 for each of the 18 sub-regions.** Habitat kms for the 18 sub-regions were computed from a glacier projection model for the years 2050 (yellow) and 2100 (green), based on 10-year averages centred around these years. The 18 sub-regions are listed from higher to lower latitude. **a** Projected kms of spawning and juvenile rearing habitat (0–2% gradient) from salmon-accessible (<10% gradient) streams larger than first order. **b** Projected kms of spawning and juvenile rearing suitable habitat (0–4% gradient) from salmon-accessible streams with steeper accessibility (<15% gradient). Uncertainty corresponds to ensemble-mean ± one standard deviation and originate from: GCM projections (for RCP4.5; $N = 5$), ice thickness estimates, and stream segment length (see Methods section "Uncertainty estimates"). Projected future salmon-accessible stream kms (±one standard deviation) are presented in Source Data File.

contrasting impacts on downstream salmon habitats across an extensive region via a variety of mechanisms.

Glacier retreat can also be associated with an increase in natural hazards that are difficult to predict, such as glacier outburst floods[40], landslides[41], or river piracy[42]. These stochastic processes can exert strong local controls on watersheds, river habitats, and salmon populations[43], and may reset the general successional trajectories of stream channel evolution following glacier retreat[44]. In addition, adult spawning salmon stray, rapidly colonize, and expand into new habitats[45] and thus rates of salmon population expansion in new streams can vary depending on the size and distance to source populations[12]. River habitat evolution, such as channel stabilization and decreased turbidity, can also take additional time in order for newly accessible and appropriate salmon habitat to become more productive for salmon[9–12]. Thus, although our study reveals the magnitude of future gains in salmon habitat, the processes of habitat creation salmon colonization, and salmon population expansion are not deterministic and will be highly influenced by local geologic and climatological processes.

More broadly, rapid contemporary glacier retreat is only one consequence of anthropogenic climate change; the realized effects of glacier retreat on Pacific salmon populations will depend on interactions with other climate-induced stressors such as ocean heat waves[46], ocean acidification[47], sea-level rise, and its effect on coastal habitats, warming air temperature, and extreme flood events or droughts, all of which could cause widespread declines in salmon abundance[3].

Understanding future shifts in suitable habitat for Pacific salmon and other species of importance[48] can support forward-looking management and conservation. For example, the heavily glacierized 'transboundary region' of southeast Alaska/British Columbia/Yukon, which has substantial forecasted gains in salmon habitat, is also concurrently experiencing a modern-day gold rush[16]. Mineral claims have been staked in regions currently covered by ice, and mines have been approved in recently deglacierized areas. Effective protection of Pacific salmon will entail conserving not just their current habitat, but also avoiding the degradation of their future habitat. Whether Arctic drainages or glacier-covered watersheds, there is an urgent need for science to inform the conservation and management of Earth's climate frontiers for the climate resilience of Pacific salmon and other species of importance[49–52].

## Methods

**Sub-regions**. The study region focuses on 18 sub-regions within the Pacific mountain ranges of North American overlapping with the range of Pacific Salmon with >1.5% glacier cover (Figs. 1 and 2). The term "sub-region" here refers to either a single major salmon watershed or aggregates of small coastal watersheds, which range in area from ~13,000 to ~68,000 km². For sub-regions within Alaska, USA, we accessed boundary data from the Watershed Boundary Database at the USGS (https://www.usgs.gov/). For sub-regions within British Columbia, Canada, we accessed boundary data from the Freshwater Atlas of British Columbia (https://catalogue.data.gov.bc.ca/). Pacific salmon range data were from the National Center for Ecological Analysis and Synthesis (Fig. 1). The study region covers ~623,000 km² across British Columbia, Canada and Alaska, USA and ~20% of the total North American range of Pacific salmon.

**Glacier outlines**. Outlines for the 45,963 glaciers within the study region were obtained from the Randolph Glacier Inventory v6.0 (https://www.glims.org/RGI/; RGI v6.0), which provides a globally complete data set of glacier outlines outside of Greenland and Antarctic ice sheets[17]. These glaciers cover a total area of ~81,000 km², which corresponds to 80% of the total glacier area in the Pacific mountain ranges within North America. The glacier outlines refer roughly to the years 2009 ± 2 for Alaska, and 2004 ± 5 for Western Canada[17,53]. Glacierization for each of 18 sub-regions ranges from 1.5 to 52%.

**Present-day streams**. Synthetic stream networks were constructed from Digital Elevation Models (DEMs) for each of the 18 sub-regions using Geographic

Information Systems (GIS; ArcGIS 10.6 and QGIS 2.18) hydrology tools to represent present-day streams throughout the study region. Specifically, we used Advanced Spaceborne Thermal Emission and Reflection Radiometer (ASTER) global DEMs v2.0 with a spatial resolution of ~30 m[54]. Open access synthetic stream network datasets such as the National Hydrography Dataset (NHD) from the USGS and the Freshwater Atlas from the British Columbia government are available but were not used due to inconsistencies in spatial resolution across the study region. From our synthetic stream networks, we eliminated all stream segments that overlapped with the RGI glacier outlines because the ASTER global DEMs used to create the synthetic stream networks represent glacier surface elevation rather than estimated deglaciated terrain. All present-day streams within our study region are void of any major dams that inhibit salmon movement based on existing databases of dams[55]. To summarize present-day stream kms, and all subsequent analyses, we used rstudio: 1.4.1103-4, R: 'Mirrors'.

**Identifying and verifying stream gradient thresholds for migrating salmon and for determining accessible glaciers**. We used stream gradient-based thresholds the determine constraints in salmon migration and the number of glaciers that would be accessible and create future streams for migrating adult salmon. Based on the large body of literature suggesting stream gradients (e.g., ranging from <10–20%) suitable for migrating Pacific salmon[25–27,29], and our own validation results (see below), we applied a conservative stream gradient threshold of 10%, and more inclusive alternative of 15%, to the synthetic present-day stream networks representing streams suitable for migrating adult salmon. The salmon migration constrained synthetic present-day stream network was used to identify which glaciers would be accessible to adult salmon. The 15% stream gradient threshold represents accessibility for salmon that are capable of swimming up steeper gradients (e.g., Chinook salmon). We calculated stream gradient within the present-day stream network by breaking the network into ~500 m segments, then determining the slope of each stream segment by extracting elevation values from the ASTER global DEMs for both ends of each segment, then dividing the elevation difference by the segment length. The ~500 m stream segment length was the minimum distance possible given the spatial resolution of our study region and memory limitations with the ArcGIS software (see Uncertainties). The upper limits of salmon migration within present-day streams were identified by selecting contiguous stream segments in the direction from the river mouth to headwaters (i.e., glacier tongue) that were below the different stream gradient thresholds (10 and 15%).

The migration stream gradient thresholds selected for this analysis were determined based on our validation analysis and those presented in the literature. For the validation analysis, we used Pacific salmon location data from the Anadromous Waters Catalogue (AWC; www.adfg.alaska.gov/sf/SARR/AWC) and segment slope values from our synthetic present-day stream network, for a sample watershed, the Susitna River watershed. The AWC is maintained by the State of Alaska and contains observation data for all Pacific salmon species with associated spatial reference (latitude and longitude). However, given the difficulties inherent in surveying for fish throughout Alaska, the AWC is incomplete, and Pacific salmon are likely to present in many unsurveyed streams. Thus, our analysis of fish presence within streams represents a minimum of the extent of fish occurrence and therefore is conservative. We used the Susitna River watershed for our validation analysis given that it supports good coverage of all five species of Pacific salmon, with each species having a minimum of ~200 individual observations (Supplementary Fig. 1). In addition, the Susitna River watershed is large in size (Supplementary Table 1), containing a range of watershed types (i.e., rain, snow, and glacier dominated) representing diversity in salmon habitats[56].

We ran the GIS network analysis tool to assess the maximum stream gradient each Pacific salmon could migrate beyond. Using our present-day stream network (segmented into ~500 m stream lengths containing slope values), we obtained the maximum stream gradient value crossed by each salmon species (obtained from the AWC) from river outlet to observation location (Supplementary Fig. 1). For each salmon species, apart from coho, ~75% of the salmon observations cross a maximum stream gradient threshold of at least 10%, with Chinook, chum, and pink salmon only infrequently being observed in stream segments beyond ≥10% (Supplementary Fig. 1). Given the migration thresholds identified in the literature[25–27,29] and this verification analysis, we confirmed that the migration stream gradient thresholds of 10% represent a conservative estimate, and 15% represent a more inclusive estimate. Our analysis covering 623,000 km² should be viewed as the best-available predictions that require field validation at specific locations.

After determining the salmon migration thresholds, we selected the glaciers that were butted against the salmon migration constrained present-day stream network using the 10% and 15% thresholds. In other words, the selected glaciers that feed into streams that have the potential to currently be colonized by migrating salmon. Thus, as the glaciers retreat, they present new habitat that salmon can colonize. We define these glaciers as "accessible".

**Future salmon-accessible streams created from estimated deglaciated bedrock**. From each of the accessible glaciers, we created future stream networks from estimated deglaciated bedrock. We estimated the deglaciated bedrock terrain by subtracting gridded ice thickness data from ice surface DEMs for every accessible

glacier within our study region. Ice thickness distribution was calculated at a grid resolution of 25–200 m (depending on glacier area) using a simple dynamic model that considers glacier mass turnover and ice flow mechanics, and by inverting the glaciers' surface topography[57] (described below). The data set uses glacier outlines from RGI v6.0 and is in close agreement with the recently released global consensus glacier ice thickness product (see Methods)[58]. Additionally, we compared the ice thicknesses and overall ice volumes used in this analysis[59], with a more recent ice thickness data set[58], and our assessment indicates that the differences are small (<3%) with respect to the other uncertainties that we account for. Inferred ice thickness was validated against a set of ice-penetrating radar observations for 300 glaciers from most glacierized regions of the world[57]. Ice surface DEMs were obtained from the Shuttle Radar Topography Mission DEMs for glaciers below 60°N (resolution of ~90 m), and from ASTER global DEMs for glaciers above 60°N (resolution of ~30 m). Both DEMs had an elevation uncertainty of ~±10–20 m for mountain areas[57].

We built a future stream network derived from estimated deglaciated bedrock terrain beneath accessible glaciers using the ArcGIS hydrology toolbox. We extracted slope values from each ~500 m segment using the same methods as used for the present-day stream network. To each segment, we assigned a stream order to each stream segment using the program RivEX to determine each stream size. Stream order is a metric used to measure the relative size of streams, where the smallest tributaries are referred to as first-order streams, which flow into larger streams that combine to form streams of higher order. All future stream segments derived from deglaciated bedrock terrain beneath accessible glaciers below the adult migration thresholds (<10 and 15%) of all sizes (stream orders First–Fourth in this case) were termed "future salmon-accessible streams". The absolute future salmon-accessible stream kms were summed for each of the 18 sub-regions (Supplementary Table 3) to determine the extent of new streams suitable for migrating adult salmon. Last, we calculated the relative increase in stream kms for each of the 18 sub-regions by dividing the total future salmon-accessible stream kms, projected to be created in 2100, by the total present-day stream kms below either 10% (Fig. 2) or 15% (Supplementary Fig. 2) stream gradient threshold.

**Glacier retreat modeling and exposure of future salmon-accessible streams**. To project the retreat of glaciers, we applied the Global Glacier Evolution Model (GloGEM) to each of the accessible glaciers that requires the use of DEMs and glacier outlines (RGI v6.0)[23]. GloGEM computes glacier mass balance and associated geometry changes for each individual glacier in the study region[23]. To calculate glacier surface mass balance, as a difference between accumulation (snowfall and refreezing) and ablation (glacier surface melting), GloGEM was forced with a monthly time series of near-surface air temperature and precipitation. Glacier geometry changes (e.g., thinning and/or shrinking) were assessed from the surface mass balance coupled with the empirically derived functions of glacier thinning along the glacier centerline[60]. For annual mass losses at marine- or lake-terminating glacier fronts, glacier retreat was approximated by accounting for glacier front height and width[61]. The total mass changes for each glacier were used to adjust surface elevation and extent on a yearly basis. Previous work gives further details of the model, its calibration, and downscaling procedures[23,39].

To project the glacier retreat for the benchmark years 2050 and 2100, we forced the glacier model with temperature and precipitation time series from an ensemble of five GCMs. The five GCM models were selected as they showed better performance in simulating climatology over North America relative to other Coupled Model Intercomparison Project 5 GCMs:[62] CanESM2, CSIRO-Mk3-6-0, GFDL-CM3, MIROC-ESM, and MPI-ESM-LR. The GCMs are subjected to a range of specified climate forcings that correspond to plausible scenarios for the rate of change in the concentration of atmospheric $CO_2$ and other greenhouse gases. For the IPCC AR5, these scenarios are referred to as Representative Concentration Pathways (RCPs) and the standard emission scenarios are RCP2.6, RCP4.5, RCP6.0, RCP8.5[63]. For several of the GCMs we used, the RCP6.0 was omitted, and the RCP2.6 is likely to not be reached[64], therefore we selected the RCP4.5 and RCP8.5 for the glacier modeling. The glacier model is forced by each GCM in the ensemble, whereas the projections of glacier retreat are presented as the ensemble mean for each RCP.

We used the glacier retreat projections to assess when each future salmon-accessible stream segment, below either the 10% or 15% stream gradient thresholds, would become exposed from glacier ice based on 10-year averages of modeled glacier extent centered around the years 2050 and 2100 for both the RCP4.5 and RCP8.5 and each of the five GCMs (Fig. 2 and Supplementary Fig 2), and the ensemble-mean. We then summed the stream kms that would be available by the projected 2050 and 2100 years for each of the 18 sub-regions (Supplementary Table 3), and the total stream kms by stream order after complete deglaciation (Supplementary Table 6).

**Defining salmon habitat requirements using stream gradient and order**. Based on a strong body of literature on habitat suitability (Supplementary Table 2), we determined how many of the future salmon-accessible stream kms could be used specifically for salmon spawning and rearing habitat by selecting stream segments with gradients using two scenarios: either a 0–2% or 0–4%. We also define salmon spawning and rearing habitat as having a stream order greater than first order.

Although we acknowledge that salmon can use first-order streams, we focus on streams greater than first order, which includes second-, third-, and fourth-order streams. Thus, our analysis indicates that suitable spawning and rearing habitat from the future salmon-accessible stream networks had segments with stream orders that ranged from second to fourth, with stream gradients ranging from either 0–2% or 0–4%.

From the future salmon-accessible streams, we determined the amount of salmon habitat by summing the stream kms based on two scenarios (i.e., 0–2% spawning/rearing with a < 10% stream gradient threshold; and, 0–4% spawning/rearing with a <15% threshold), then determined when the stream habitat would become available for the years 2050 and 2100 for each of the 18 sub-regions (Fig. 3). The two scenarios help capture the fact that different salmon species have different tendencies in terms of stream gradients associated with spawning and rearing (Fig. 3)[32].

**Uncertainty estimates**. Given that our study integrated models, data inputs, and analytical approaches for determining future salmon habitat created by glacier retreat across a large study region, it is important to consider potential uncertainties. All values presented throughout the manuscript with uncertainty estimates are propagated from the uncertainty derived from the (1) GCM projections, and from the mean uncertainty extracted from the results of our sensitivity analyses of (2) glacier ice thickness estimates, and (3) stream segment length, as presented below. We used the root sum of squares from the three individual uncertainty estimates to determine the global uncertainty. Uncertainty estimates are presented throughout the paper as ± one standard deviation. Given the time and computation power required to run some of these sensitivity analyses, we use sample sub-regions, North Southeast, AK, and Taku River, BC, that represent diversity in watershed size, terrain complexity, and extent of future stream network then applied the error to other sub-regions (Supplementary Table 1).

Other sources of uncertainty that were considered and discussed below, but not included in our uncertainty estimates, are the (4) glacier retreat model (GloGEM) used in our analysis, and (5) Unknown potential changes in habitat and landscape variables important to salmon spawning and rearing.

*GCM projections*. There are uncertainties in the IPCC GCM that project future greenhouse gas emissions, which appears to be a dominant source of error when considering rates of glacier retreat[19,24]. However, climate models similar to those used in this analysis have shown to be quite accurate at predicting forecasted temperature changes[65]. To illustrate the uncertainty of the climate models, for each of the 18 sub-regions, we present the individual five GCMs projections as one standard deviation around the ensemble mean.

*Glacier ice thickness estimates*. There is considerable uncertainty in modeled ice thickness distribution. An intercomparison of ice thickness models showed a ±25.9% uncertainty in inferred ice thickness[58]. Thus, for our analysis, there might be additional uncertainty due to the estimated bedrock elevation in the deglaciated topography. However, ice thickness models show a good performance regarding the patterns of thickness distribution (thin/thick parts of the glacier), even when the average estimated ice thickness of an individual glacier may be too high or low. Moreover, errors in estimated elevation of the bedrock are likely similar at either end of a ~500 m segment[59]. Hence, the errors in the calculation of slopes are likely reduced due to the spatial correlation of errors in the ice thickness estimates.

To determine the uncertainty of the future stream kms derived from deglaciated bedrock topography, obtained by subtracting ice thickness from ice surface DEMs, we ran a sensitivity test on the ice thickness data. In a conservative approach, we systematically increased/decreased all ice thicknesses by 25.9% according to the uncertainty stated in Farinotti et al. (2019) and re-computed bedrock topography. We then re-ran the ~500 m segment analysis to determine stream gradient, and applied the salmon migration constraints of 10%, and calculated the total number of future stream kms. We applied this analysis to the sample watersheds, North Southeast, AK, and Taku River, BC. Our sensitivity analysis resulted in estimates of total future stream kms for these sub-regions that were 13% greater (given error in increased ice thickness) or 12% less (decreased ice thickness) than our predictions. Thus, the error associated with the ice thickness data used to determine total future stream kms is comparable to the error from uncertain climate forcing for a given $CO_2$-emission pathway presented in the main text.

*Stream segment length*. The stream gradient thresholds applied were derived from ~500 m segment lengths and a DEM with a 30 m resolution. We selected a ~500 m segment length as it was not feasible due to limited computation resources to shorten our segment length. However, there is some imprecision in using a ~500 m segment length, such as it is impossible to know the exact stream characteristics present (e.g., longer series of riffles vs. a single large waterfalls), and Pacific salmon upstream migration is generally restricted by certain stream features such as waterfalls. In addition, we could not validate our stream gradient thresholds for adult salmon migration against known barriers (e.g., waterfalls) because there are no known data sets of salmon migration barriers between southern British Columbia and Alaska. We also acknowledge that some migration barriers can vary seasonally depending on stream flows. Therefore, some segments with a 10%

stream gradient may contain a migration barrier whereas others may not. For comparison, other studies have used similar reach lengths of 200 m[66] and 500 m[67].

To assess the uncertainty arising from the assumptions on the segment lengths used throughout the study, we ran two sensitivity analyses, (1) on the present-day stream network, which determines the number of accessible glaciers, and (2) on the future stream kilometres. First, we broke the present-day salmon migration stream network into different segment lengths (~250 m, ~400 m, ~600 m, and ~750 m segment length with a 10% stream gradient threshold), as the number of accessible glaciers is determined by the present-day stream network. We applied this sensitivity analysis on the two sample sub-regions, North Southeast, AK, and Taku River, BC. For the North Southeast, AK sub-region the number (and size) of salmon-accessible glaciers changed within the range of −16 to 9% (~90–~350 ha), depending on the chosen segment length (Supplementary Table 7). For the Taku River, BC sub-region there was the same number of salmon-accessible glaciers in all segment length scenarios (Supplementary Table 7). Second, we estimated the uncertainty of the future stream networks segment length derived from each accessible glacier's future deglaciated terrain (see below). For this analysis, we used a ~250 m and ~750 m segment lengths, and 10% stream gradient threshold, for each of the 18 sub-regions to understand variations in total future salmon-accessible stream kms given different segment lengths (Supplementary Table 8). On average for all 18 sub-regions, the total future salmon-accessible stream kms when using the ~250 m segment length was 14% less, and when using the ~750 m segment length, was 11% more. The ~250 m segment length is more precise and therefore there are more opportunities for the slope calculation to be above the 10% stream gradient threshold, whereas the opposite is true of the ~750 m segment length. In addition, the stream network using the ~750 m segment lengths extends further into the upper reaches of the stream network, leading to more salmon-accessible stream kms. Given that for some glaciers the spatial resolution of the ice thickness data was only 200 m, that there were computational limitations, and to be consistent with the present-day segment lengths for comparison purposes, we chose to use the ~500 m segment length, a good trade-off between capturing the accuracy and precision of the stream network. The associated error in determining segment length is less than or equal to the error in using the different GCMs as presented in the main findings (Supplementary Table 3).

*Glacier retreat model.* We predicted rates of glacier retreat using GloGEM, as described above. A recent intercomparison of global glacier models showed that GloGEM predicts somewhat faster rates of glacier retreat compared with some other models[19]. However, the most recent GloGEM runs, based on a new glacier-specific data set of mass balances 2000–2020[20], are consistent with the results used in the present study for the two relevant regions Alaska and Western Canada. This indicates that the model correctly captures glaciers changes in the past and is, thus, likely to yield reasonable projections for the next decades.

*Unknown potential changes in habitat and landscape variables important to salmon spawning and rearing.* Many habitats and landscape variables (e.g., channel width and confinement, streamflow, stream temperature, riparian forest development) can be important to Pacific salmon throughout their life cycle[29,68–71], but it was not possible to consider these explicitly in this study. Field measurements of variables such as sediment supply, grain size, bankfull discharge, and channel slope are not readily available over large geographic areas[72,73], and are impossible to obtain for sub-glacier environments. Many studies have determined ways to extract important environmental variables for salmon using DEMs[66,74]. However, this type of analysis was not possible given the uncertainties and errors in estimating the deglaciated bedrock topography (see above). Therefore, only stream order and gradient were used in determining salmon streams and suitable habitats for spawning and rearing[75]. These metrics are useful and accurate in identifying suitable habitats for Pacific salmon, as shown by other studies[75–77]. Further, we were not able to incorporate natural hazards associated with glacier retreat[78], such as landslides or river piracy[42], into predictive models, but acknowledge that such stochastic events can alter the accessibility and suitability of rivers following glacier retreat.

**Reporting summary**. Further information on research design is available in the Nature Research Reporting Summary linked to this article.

## Data availability

The newly accessible stream kilometer data generated in this study have been deposited in a Zenodo database[79]. All other spatial data sets were obtained from open access sources: DEMs used in this analysis can be downloaded from NASA (https://asterweb.jpl.nasa.gov/gdem.asp); watershed boundary data for the USA are available at USGS (https://www.usgs.gov), and from at Freshwater atlas of BC for British Columbia (https://www2.gov.bc.ca/gov); glacier outline data can be found at the Glacier Inventory v6.0 (https://www.glims.org/RGI/); Pacific salmon presence data are available from the AWC (www.adfg.alaska.gov/sf/SARR/AWC). Source data are provided with this paper.

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

## Acknowledgements

We gratefully acknowledge funding from the Gordon and Betty Moore Foundation for the Salmon Science Network Initiative, which provided the opportunity to hold a working group of scientists from Canada, the United States, and the United Kingdom in November 2017. Kara Pitman was supported by National Science and Engineering Research Council and Association of Canadian Universities for Northern Studies. We are extremely grateful for Valentina Radić's contribution to project design and for contributing a thorough review prior to manuscript submission.

## Author contributions

K.J.P., J.W.M., M.R.S., D.W., E.W.H., D.E.S., A.M.M., R.B., G.R.P., G.H.R. conceived the project during a working group on Pacific Salmon and Glacier Retreat held in November 2017. K.J.P. led the project, gathered, and ran an analysis of the data. M.H. provided data and ran the analysis. D.C.W. ran the analysis. T.J.B. provided insights on analysis. K.J.P. and J.W.M. drafted the initial manuscript. All authors contributed, provided input, and approved the text in the manuscript.

## Competing interests

The authors declare no competing interests.

## Additional information

**Peer review information** *Nature Communications* thanks Ben Marzeion, Keith Nislow, and the other anonymous reviewer(s) for their contribution to the peer review this work. Peer reviewer reports are available.

