## [Peer Review File · Nature Communications]

Reviewers' Comments:

Reviewer #1:

Remarks to the Author:

Pitman et al. investigate the impact of the deglaciation of Western North America under anthropogenic climate change on the habitat of Pacific salmon. The study is not only relevant as an assessment of this specific future ecosystem change, but also an example of how increasing complexity of the models used to project climate and its impact in the Earth system allows new interdisciplinary studies, that previously would have been hampered by a mismatch between the level of detail needed on one side (e.g., habitat modeling), and possible to provide on the other side (e.g., glacier projections). Because of this wider relevance, the study is well suited for a journal like Nature Communications.

Because of the interdisciplinary nature of the study, I am not able to assess all aspects of the material. Specifically, I am not familiar with the state-of-the-art concerning habitat modeling, and generally the ecosystem part of the study. I therefore focus my attention on the glacier projection part.

The paper is very well written and easy to follow also for the non-specialist. The figures are well-designed, but some of the labels seem very small and are hard to read without significant magnification on the screen. As far as I can assess, the conclusions of the paper – a significant increase in the length of streams accessible to salmon – are robustly backed by the evidence presented by the authors. The methods are explained in enough detail to ensure reproducibility.

I only have two general comments that I think should be considered by the authors and the editor before the manuscript is eventually accepted for publication. Both concern the uncertainty assessment of the results, and while I don't see the conclusions threatened by any of them, they might lead to a widening of the error margins – or at least a qualitative discussion of additional errors that are hard to quantify.

(1) The glacier model used for the projections, GloGEM, is state-of-the-art and well suited for a study like this, given that among the glacier models able to be applied at the relevant scale, it provides one of the more realistic representations of glacier geometry change. However, in a recent intercomparison, GloGEM tended to produce more negative mass balances, and specifically a faster glacier area retreat than the ensemble median in the regions of interest ("Alaska" and "Western Canada and US" in the supplement of DOI: 10.1029/2019EF001470). This will impact the timing of streams becoming accessible to salmon in the future, potentially overestimating the effect. This is probably not even a dominant uncertainty of the manuscript's results, but one that should be mentioned in the discussion. Potentially, the authors may consider a very simple estimate of the impact by assuming a very rough time shift of 20 years or so of the projected glacier areas, which is roughly the difference between the GloGEM projections and the ensemble mean projections of the relevant regions in the intercomparison.

(2) I agree with the assessment that errors of the ice thickness estimate are probably more relevant on the glacier-wide scale than on the scale of segments, such that they don't matter much for estimating the slope of individual segments. However, since the ice thickness estimates are constrained to zero at the glacier boundary, a positive error in the overall glacier thickness would lead to a "too convex" glacier, implying too low bedrock slopes at the lower portions of the glacier. A negative error in the overall glacier thickness would lead to too steep bedrock slopes at the lower portion of the glacier – but this case is constrained by the (known) steepness of the ice surface. I.e., I'm wondering whether it is justified to assume that the error distribution in the slope of the bedrock is symmetrical (and thus cancels if the sample size is large enough), since it is constrained on one side (the ice surface), but not the other (the unknown bedrock). I guess that this will also depend on the relation of the slope of the ice surfaces to the relevant slope thresholds, and I'm not at all sure how to address this. But maybe the authors have already thought along these lines and could include a brief discussion in the Uncertainties section, or perhaps they can rule out any significant impact on the uncertainties.

Ben Marzeion

Reviewer #2:

Remarks to the Author:

In this study, the authors aim to estimate the potential increase in habitat of Pacific salmon linked to the glacier retreat in the Pacific mountain ranges of western North America. The study is based on the combination of a geographical and a glaciological analysis. Future glacier retreat was estimated based on the present glacier outlines (RGI), DEMs and using a simple dynamic glaciological model (GloGEM). The glaciological model was forced with temperature and precipitation projections from 5 Global Climate Models for the two climate emission scenarios RCP 4.5 and 8.5. Glacier retreat projections were estimated for the years 2050 and 2100.

Deglaciated bedrock terrain elevation models were estimated by removing the RGI glacier outlines from the DEMs for the present day, and by subtracting the estimated ice thickness distribution from the DEMs for the years 2050, 2100.

Stream networks were then constructed based on those deglaciated bedrock terrain elevation models, and Pacific salmon habitats defined as stream reaches with continuous < 10% (or <15%) stream gradient from the river mouth.

I found the idea of this study really very interesting. The main objective of assessing the potential gains in future Pacific Salmon habitat with the purpose to inform on ecosystem protection is indeed of particular importance.

However, I found it difficult to evaluate the relevance of concluding values given the multiple-level uncertainties.

Although the authors mentioned a list of sources of uncertainty, there were not all quantified and integrated along the analysis chain. I agree it would be complicated, if not impossible to produce relevant uncertainties of new accessible stream length. However, it would be necessary to provide (and/or discuss) more clearly uncertainties at the different steps. While the majority of the analysis arises from the glaciological models, providing quantitative measures with uncertainties, the ecological conclusion of this study seems rather qualitative, with few and still too unclear information to judge the order of magnitude provided.

I acknowledge the authors could not provide a global uncertainty on new stream habitat length but more information/values on uncertainties could be presented and discussed at the different levels, spatial, and temporal scales. After reading the entire manuscript, I was not sure how to deal with the concluding value (without any range) proposed. ~6000 km : an order of magnitude? A minimum of ? 6000 km among how many km within the entire study region? What does this value depict? Sometimes information and values are provided for the entire Pacific mountain ranges of western North America, sometimes for specific regions (not necessarily the same region according to the argument highlighted). It remains thus quite difficult to have a clear idea regarding the concluding results.

Please, see my comments below and in the pdf file for examples.

Regarding the methods:

The methods would be easier to follow by improving their organisation. In particular, provide all information concerning the same topic within the same paragraph. Avoid repetition.

L 323. We selected glaciers that were within 100 m of the present-day stream network below adult salmon migration stream gradient thresholds of 10% and 15%. We define these glaciers as "accessible".

This is a major decision point in the study. And, I do not understand why you selected a threshold to exclude accessible glaciers and how you selected this 100m value. This choice has profound impact on the results, excluding a considerable number of glaciers, thereby of new stream networks, and should thus be justified/documentated.

Regarding the selection of the < 10% (or <15%) threshold:

I acknowledge you first selected estimates from the literature but the verification on the study

region based on the Anadromous Waters Catalogue remained unclear and/or incomplete to me.

In your analysis (extended Data Fig. 1), you used the median stream gradient along stream segments from stream outlet to point data. I am not sure the median value is the most relevant index. Anyway, it might be good to explain/justify it.

In addition; How, in the extended Data Fig 1, the upper quartiles of the boxplot showing median stream gradient [...] could represent the maximum gradient each Pacific salmon species could access?

L310. You rather mentioned 'the maximum stream gradient that salmon could pass'

Also; Because you have various data points along the same stream segment, downstream segment values are included various time => how does this affect the understanding of the salmon distribution?

After selecting this threshold, did you calculate how many kms of stream with the required conditions (< 10 or 15%) were not colonised by salmon (with no salmon occurrence). This would allow identifying other factors, such as proximity to glacier, % in glacier cover in the catchment or others.

Why did you perform the verification only on the Susitna River basin? It would be very interesting to perform this exercise in various/all watersheds with available data, including for assessing the variability across catchments, identifying other factors, such as the effects of glacial influence, altitude, latitude... Even if only based on the Anadromous Waters Catalogue (no exhaustive database), this region-scale analyse would provide a first idea of pacific salmon distribution, also estimate the effect of glacier influence on the different species. Has this been performed earlier? It might be good beginning with this.

Regarding the Uncertainties.

To evaluate the impact of segment length, I would rather focus in examining the effect on the size of the stream network instead of on the number of accessible glaciers, which minimises/blurs the impact of the segment length.

In the Results/Discussion:

In general, the discussion could be improved, shortening some parts, removing repetitions. Several times, two consecutive sentences provide the same information, with one including values the other without values. Certain sentences provide the same information as in the introduction.

A major point is that, even if listed, uncertainties could not be quantified. Nevertheless, the habitat estimations could be better presented and discussed to allow the reader evaluating the concluding results.

There is a lack of a comprehensive summary, presenting the main results in their context. Indeed, results are presented either at the global scale (entire region) or at the sub-catchment scale, using either percentages or absolute values, often without mentioning the reference absolute values (e.g., X new stream habitat kms among ??? kms, X accessible glaciers among ??? glacier). It will be very useful to have a synthetic table with for the 18 sub-regions and the entire study region: the catchment size (km²), nbr of glaciers, glacier area (km²), stream network size (km), nbr of accessible glaciers, future stream network size (km)...

Throughout the manuscript, you insisted on the conservative options you selected. For example, L161-164: 'We conservatively estimate that glacier retreat will create a total of ~1,900 kms of future salmon [...]

I would not be so affirmative and would soften the tone of the estimations.

I believe it is important to mention and develop that salmon might not colonise just after glaciers have gone, and to insist on the potential delay between glacier retreat and salmon establishment, especially as you are interesting in the timing of emerging salmon habitats.

Another point to mention/develop would be the non-natural barriers (dams...).

I also believe it is crucial to develop the discussion on the importance of flow, temperature and glacial influence on salmon as well as the impacts of glacier retreat on these factors although those parameters could not be included in your models.

Please also see the pdf for more specific (not necessarily minor) comments.

Reviewer #3:

Remarks to the Author:

This manuscript presents the results of a study linking climate change, glacial retreat, and increases in potential Pacific salmon habitat. Combining climate model-driven estimates of glacial loss and physical thresholds for salmon migration, spawning and rearing, the authors conclude that glacial retreat could open up a large amount of potential habitat with the potential to provide compensation for losses elsewhere with the current geographical range of Pacific salmon. While the idea of northern distributional expansion has been much discussed, and in some cases documented in the general ecological literature, this is one of the few studies to quantify the likely magnitude of the effect and link it to a specific climate-driven mechanism. Given this novelty, general potential implications, and the ecological, economic, and cultural importance of salmon, I think that the manuscript meets the standards and requirements of the journal. I found it clear and easy to follow, and have only a few relatively minor comments:

1. While some reference is made to the generality of potential northern expansion across species and systems, I still found the paper to be a bit salmon-centric for Nature and encourage the authors to think more broadly.
2. The authors make a very good point about the need to think pro-actively in terms of conservation and protection of these new habitats (which could again be a more general consideration across many species for northern expansion). I wonder if the results also suggest the particular importance of the adjacent source populations and their habitats in providing colonists.
3. I understand that the authors cannot include all the complex details of channel evolution following glacial retreat into their models and analyses. However, I was curious about the relationship between channel width (with its associated influence on total habitat area) and the habitats provided by glacial retreat. If new channels tend to be narrow, while existing habitat includes quite a lot of wide channels further down in the drainage network, how will this affect the relative importance of new habitat (essentially will the proportional contribution of new habitat be lessened if area not length is the dimension considered)?

REVIEWER COMMENTS

Reviewer #1 (Remarks to the Author):

Reviewer 1, comment 1: Pitman et al. investigate the impact of the deglaciation of Western North America under anthropogenic climate change on the habitat of Pacific salmon. The study is not only relevant as an assessment of this specific future ecosystem change, but also an example of how increasing complexity of the models used to project climate and its impact in the Earth system allows new interdisciplinary studies, that previously would have been hampered by a mismatch between the level of detail needed on one side (e.g., habitat modeling), and possible to provide on the other side (e.g., glacier projections). Because of this wider relevance, the study is well suited for a journal like Nature Communications.

Because of the interdisciplinary nature of the study, I am not able to assess all aspects of the material. Specifically, I am not familiar with the state-of-the-art concerning habitat modeling, and generally the ecosystem part of the study. I therefore focus my attention on the glacier projection part.

The paper is very well written and easy to follow also for the non-specialist. The figures are well-designed, but some of the labels seem very small and are hard to read without significant magnification on the screen. As far as I can assess, the conclusions of the paper – a significant increase in the length of streams accessible to salmon – are robustly backed by the evidence presented by the authors. The methods are explained in enough detail to ensure reproducibility.

***Response:** Thank you for the positive feedback and review. We have gone through each of the manuscript figures and increased text size.*

Reviewer 1, comment 2: I only have two general comments that I think should be considered by the authors and the editor before the manuscript is eventually accepted for publication. Both concern the uncertainty assessment of the results, and while I don't see the conclusions threatened by any of them, they might lead to a widening of the error margins – or at least a qualitative discussion of additional errors that are hard to quantify.

The glacier model used for the projections, GloGEM, is state-of-the-art and well suited for a study like this, given that among the glacier models able to be applied at the relevant scale, it provides one of the more realistic representations of glacier geometry change. However, in a recent intercomparison, GloGEM tended to produce more negative mass balances, and specifically a faster glacier area retreat than the ensemble median in the regions of interest (“Alaska” and “Western Canada and US” in the supplement of DOI: 10.1029/2019EF001470). This will impact the timing of streams becoming accessible to salmon in the future, potentially overestimating the effect. This is probably not even a dominant uncertainty of the manuscript's results, but one that should be mentioned in the discussion. Potentially, the authors may consider a very simple estimate of the impact by assuming a very rough time shift of 20 years or so of the projected glacier areas, which is roughly the difference between the GloGEM projections and the ensemble mean projections of the relevant regions in the intercomparison.

***Response:** The reviewer brings up the important consideration that different glacier models provide somewhat different projections. Specifically, the reviewer is correct that for the two relevant regions, Alaska and Western Canada, GloGEM (the glacier model used in the present*

study) projects rates of future area change higher than the median. However, projected glacier volume (see Fig. S18 and S20 of the mentioned paper: DOI: 10.1029/2019EF001470) for the two climate scenarios used here is exactly with the median for Western Canada and Alaska, and only diverges after about 2060. This seems to suggest that the structure of GloGEM accounting for the local topography of each glacier and specific feedback mechanisms is relevant to explain the differences. Only two other global glacier models in the intercomparison project (GlacierMIP2, Marzeion et al., 2020) account for the local terrain's geometry in a similar way whereas most models in the intercomparison are more parameterized. Also given that GloGEM has been calibrated to match the observed regional glacier mass changes in the past, the observation of faster area-change rates than the median of other models does not directly indicate that the speed of deglaciation is overestimated. The most recent model runs with GloGEM, based on a new glacier-specific data set of mass balances 2000-2020 (Hugonnet et al., 2021) are consistent with the results used in the present study for the two relevant regions Alaska and Western Canada. We have incorporated in the paper that GloGEM could be predicting faster rates of glacier retreat, and that there could be a delay in when the new rivers will be created. These changes can be found in the main text on lines 164-167 and in the Methods section on lines 536-543.

Reviewer 1, comment 3: I agree with the assessment that errors of the ice thickness estimate are probably more relevant on the glacier-wide scale than on the scale of segments, such that they don't matter much for estimating the slope of individual segments. However, since the ice thickness estimates are constrained to zero at the glacier boundary, a positive error in the overall glacier thickness would lead to a "too convex" glacier, implying too low bedrock slopes at the lower portions of the glacier. A negative error in the overall glacier thickness would lead to too steep bedrock slopes at the lower portion of the glacier – but this case is constrained by the (known) steepness of the ice surface. I.e., I'm wondering whether it is justified to assume that the error distribution in the slope of the bedrock is symmetrical (and thus cancels if the sample size is large enough), since it is constrained on one side (the ice surface), but not the other (the unknown bedrock). I guess that this will also depend on the relation of the slope of the ice surfaces to the relevant slope thresholds, and I'm not at all sure how to address this. But maybe the authors have already thought along these lines and could include a brief discussion in the Uncertainties section, or perhaps they can rule out any significant impact on the uncertainties.

Response: *In fact, we did not reflect about the non-symmetry of this uncertainty and would like to acknowledge the reviewer for this thoughtful comment. We agree that this uncertainty does involve specific geometries that will vary across the large number of analyzed glaciers. To investigate this aspect, we have performed a sensitivity analysis of increasing / decreasing ice thickness, then subtracting those values from ice surface DEM to calculate bedrock topography by the typical ice thickness uncertainty of 25.9% as stated by Farinotti et al. (2019) for a subset of representative sub-regions. On these new bedrock topographies, we recomputed longitudinal bedrock slope, and applied the 10% salmon migration threshold. The results indicated that there is an overall 13% (increased ice thickness) and 12% (decreased ice thickness) uncertainty in the future river kilometres for these two sub-regions due to uncertain bedrock elevation. We have included a description of this sensitivity analysis in the Methods section from lines 473-494 and propagated this uncertainty to all results.*

Reviewer #2 (Remarks to the Author):

Reviewer 2, comment 1: In this study, the authors aim to estimate the potential increase in habitat of Pacific salmon linked to the glacier retreat in the Pacific mountain ranges of western North America. The study is based on the combination of a geographical and a glaciological analysis. Future glacier retreat was estimated based on the present glacier outlines (RGI), DEMs and using a simple dynamic glaciological model (GloGEM). The glaciological model was forced with temperature and precipitation projections from 5 Global Climate Models for the two climate emission scenarios RCP 4.5 and 8.5. Glacier retreat projections were estimated for the years 2050 and 2100. Deglaciated bedrock terrain elevation models were estimated by removing the RGI glacier outlines from the DEMs for the present day, and by subtracting the estimated ice thickness distribution from the DEMs for the years 2050, 2100. Stream networks were then constructed based on those deglaciated bedrock terrain elevation models, and Pacific salmon habitats defined as stream reaches with continuous < 10% (or <15%) stream gradient from the river mouth.

I found the idea of this study really very interesting. The main objective of assessing the potential gains in future Pacific Salmon habitat with the purpose to inform on ecosystem protection is indeed of particular importance.

However, I found it difficult to evaluate the relevance of concluding values given the multiple-level uncertainties. Although the authors mentioned a list of sources of uncertainty, there were not all quantified and integrated along the analysis chain. I agree it would be complicated, if not impossible to produce relevant uncertainties of new accessible stream length. However, it would be necessary to provide (and/or discuss) more clearly uncertainties at the different steps. While the majority of the analysis arises from the glaciological models, providing quantitative measures with uncertainties, the ecological conclusion of this study seems rather qualitative, with few and still too unclear information to judge the order of magnitude provided. I acknowledge the authors could not provide a global uncertainty on new stream habitat length but more information/values on uncertainties could be presented and discussed at the different levels, spatial, and temporal scales. After reading the entire manuscript, I was not sure how to deal with the concluding value (without any range) proposed. ~6000 km: an order of magnitude? A minimum of ? 6000 km among how many km within the entire study region? What does this value depict?

Response: *The reviewer brings up a very good point. Based on this comment, we have performed a series of major analyses to consider uncertainties. Throughout the manuscript, when we are presenting a value, we provide the value of the model output \pm one standard deviation, which is the error derived from the different GCM projections for the RCP4.5 scenario (and RCP8.5 in Extended Data Fig. 2), combined with the uncertainty derived from sensitivity analysis of the glacier ice thickness estimates, and stream segment length. These new sensitivity analyses are now presented and discussed in the “Uncertainty estimates” section starting on line 449 and highlighted in Extended Data Tables 3 and 4. The revised manuscript now more-fully presents and discusses these sources of uncertainty and integrates the error into a global uncertainty. For comparison, we previously communicated the total number of future stream kilometers throughout the study region to be ~6000 km, and after integrating the derived uncertainty, this value is now presented as “6,146 (\pm 1,619; this and following uncertainty corresponds to \pm one*

standard deviation and originate from: GCM projections, ice thickness estimates, and stream segment length) km” as shown on line 137-139.

Reviewer 2, comment 2: Sometimes information and values are provided for the entire Pacific mountain ranges of western North America, sometimes for specific regions (not necessarily the same region according to the argument highlighted). It remains thus quite difficult to have a clear idea regarding the concluding results. Please, see my comments below and in the pdf file for examples.

Response: We have gone through the pdf and have changed (where suggested) results to highlight findings for the entire study region vs. sub-regions. In some sections, we still discuss the result by sub-region such as when highlighting which sub-regions will see more or less gains in salmon habitat, as the future change in salmon habitat by sub-region is a key finding of this paper.

Reviewer 2, comment 3: The methods would be easier to follow by improving their organisation. In particular, provide all information concerning the same topic within the same paragraph. Avoid repetition.

Response: We have strengthened our topic sentences and moved sections around within the methods to improve the organizational structure.

Reviewer 2, comment 4: On line L323: “We selected glaciers that were within 100 m of the present-day stream network below adult salmon migration stream gradient thresholds of 10% and 15%. We define these glaciers as “accessible.”” This is a major decision point in the study. And, I do not understand why you selected a threshold to exclude accessible glaciers and how you selected this 100m value. This choice has profound impact on the results, excluding a considerable number of glaciers, thereby of new stream networks, and should thus be justified/documentated.

Response: We see where there was confusion around this part of the analysis. We needed to select glaciers that butted against the present-day stream networks, to select glaciers that would be accessible to salmon. We selected these glaciers using a 100m buffer between the present-day river network and glaciers. This is due the fact that some present-day river networks terminated earlier than the glaciers tongue when we initially built the synthetic river network. This is a limitation in the way the hydrological tools (within ArcGIS) use flow accumulation to build river networks. However, to verify if this 100m threshold was appropriate or if the number of glaciers selected would change if we ran different thresholds, re-ran the selection tool in ArcGIS with a 5m, 50m, and 75m buffer between present-day river network and glaciers and obtained the same number of glaciers. Thus, there was no reason to use the 100m buffer. We have revised the text to reflect this change (lines 362-366).

Reviewer 2, comment 5: Regarding the selection of the < 10% (or <15%) threshold: I acknowledge you first selected estimates from the literature but the verification on the study region based on the Anadromous Waters Catalogue remained unclear and/or incomplete to me. In your analysis (extended Data Fig. 1), you used the median stream gradient along stream segments from stream outlet to point data. I am not sure the median value is the most relevant

index. Anyway, it might be good to explain/justify it. In addition, how, in the extended Data Fig 1, the upper quartiles of the boxplot showing median stream gradient [...] could represent the maximum gradient each Pacific salmon species could access?

Response: *We understand how this analysis lacked a cohesive explanation. First, we agree that the median stream gradient along stream segments is not the most relevant. Therefore, to better explain, we have expanded and clarified the verification analysis in the Methods section. Second, we have updated the Extended Data Fig 1 caption to reflect these changes (lines 787-797) and in the Methods section (lines 350-361).*

Reviewer 2, comment 6: You rather mentioned ‘the maximum stream gradient that salmon could pass’. Also, because you have various data points along the same stream segment, downstream segment values are included various time => how does this affect the understanding of the salmon distribution?

Response: *We have addressed the rationale for using the maximum stream gradient comment below in Reviewer 2, Comment L310 and Reviewer 2, comment 5. As for the distribution comment, the reviewer is correct, downstream stream gradient values are included numerous times given that each salmon observation location migrated through the entire river network with lower portions of the network having higher frequency. Understanding the salmon distribution is a very important question, but not relevant to our stream gradient verification analysis.*

Reviewer 2, comment 7: After selecting this threshold, did you calculate how many kms of stream with the required conditions (< 10 or 15%) were not colonised by salmon (with no salmon occurrence). This would allow identifying other factors, such as proximity to glacier, % in glacier cover in the catchment or others.

Response: *This is an interesting question, but the data are not suited for these analyses. Specifically, the AWC dataset only showcases salmon presence, salmon absence can only be inferred indirectly, and salmon observations may be compounded by high levels of turbidity associated with glacier streams. Thus, the suggested analyses are beyond the scope of the present paper. Instead, we have modified the text to highlight that this supplemental analysis was performed to explore the spatial extent of salmon within this watershed as a function of slope. We have now explained the limitations to the AWC dataset in the methods section (lines 339-343).*

Reviewer 2, comment 8: Why did you perform the verification only on the Susitna River basin? It would be very interesting to perform this exercise in various/all watersheds with available data, included for assessing the variability across catchments, identifying other factors, such as the effects of glacial influence, altitude, latitude... Even if only based on the Anadromous Waters Catalogue (no exhaustive database), this region-scale analyse would provide a first idea of pacific salmon distribution, also estimate the effect of glacier influence on the different species. Has this been performed earlier? It might be good beginning with this.

Response: *We appreciate this suggestion and we have explored the proposed approach in various forms. However, as noted above, there are severe limitations with the existing salmon dataset. One major limitation to this kind of analysis is the lack of good data on salmon*

abundance in streams that are glacier-fed given their high levels of turbidity and challenges in determining where there are fish. Due to this challenge, the salmon presence data may be incorrectly skewed, and inferences may be made regarding the lack of salmon presence in areas where there may be salmon. This is a challenge, and more research needs to be done to ensure that accurate salmon abundance within turbid waters, such as those fed by glaciers, is properly assessed. We elaborate why we selected the Susitna River watershed in our response to Reviewer2, Comment L306.

Reviewer 2, comment 9: To evaluate the impact of segment length, I would rather focus in examining the effect on the size of the stream network instead of on the number of accessible glaciers, which minimises/blurs the impact of the segment length.

***Response:** In fact, we did run a sensitivity analysis on the future stream networks in addition to the number of glaciers selected. We can not do this at the same time as the analysis process is slightly different. The number of accessible glaciers is based on where the migrating salmon constrained present-day rivers connect to glaciers. Whereas the future stream network is determined without the use of the present-day stream network. Therefore, it is a two-step process. First, we needed to determine which glaciers are accessible, then build a future network from the deglaciated terrain using the ice surface DEM and ice thickness data. We have modified the manuscript to address this confusion, and now describe in detail how each sensitivity analysis was conducted in the Methods section on lines 508-535.*

Reviewer 2, comment 10: In general, the discussion could be improved, shortening some parts, removing repetitions. Several times, two consecutive sentences provide the same information, with one including values the other without values. Certain sentences provide the same information as in the introduction.

***Response:** We have reworked the discussion to make it clearer and more concise.*

Reviewer 2, comment 11: A major point is that, even if listed, uncertainties could not be quantified. Nevertheless, the habitat estimations could be better presented and discussed to allow the reader evaluating the concluding results.

***Response:** As described above, we have performed a series of major changes to the manuscript to consider and communicate uncertainties more fully. We elaborate in the “Uncertainty estimates” section (starting on line 449) how the error was derived and present the values in the new Extended Data Tables 3 and 4.*

Reviewer 2, comment 12: There is a lack of a comprehensive summary, presenting the main results in their context. Indeed, results are presented either at the global scale (entire region) or at the sub-catchment scale, using either percentages or absolute values, often without mentioning the reference absolute values (e.g., X new stream habitat kms among ??? kms, X accessible glaciers among ??? glacier). It will be very useful to have a synthetic table with for the 18 sub-regions and the entire study region: the catchment size (km²), nbr of glaciers, glacier area (km²), stream network size (km), nbr of accessible glaciers, future stream network size (km)...

***Response:** This is a great suggestion. We have included three new tables (Extended Data Table 1,3 and 4) to address this comment.*

Reviewer 2, comment 13: Throughout the manuscript, you insisted on the conservative options you selected. For example, L161-164: ‘We conservatively estimate that glacier retreat will create a total of ~1,900 kms of future salmon [...] I would not be so affirmative and would soften the tone of the estimations.

Response: Throughout the manuscript, we have softened the tone of our projections of future salmon habitat gains. In this example, we have changed the language to “we conservatively estimate that glacier retreat will create 1,930 (+569) kms of future salmon spawning and rearing habitat (gradient threshold <10%, 0-2% gradient for spawning and rearing habitat) by 2100 under RCP 4.5 (Fig. 3) throughout the entire study region”. By incorporating uncertainty estimates into our projections, we hope that the tone of our estimates will be softened.

Reviewer 2, comment 14: I believe it is important to mention and develop that salmon might not colonise just after glaciers have gone, and to insist on the potential delay between glacier retreat and salmon establishment, especially as you are interested in the timing of emerging salmon habitats.

Response: We have revised the text within the manuscript to reflect this comment. To add, Milner et al, 2011 found that salmon colonization occurred quite quickly in a newly deglaciated stream, with salmon populations appearing within 15 years of the stream’s creation. Thus, while time could improve these salmon habitats, salmon are also really well adapted to different habitat types and can colonize quite quickly, specifically if there is a strong nearby donor population. However, in other locations, depending on sediment and water supply it may take longer for streams to colonize than the example provided (Milner et al., 2011). We have included these important details on lines 223-230.

Reviewer 2, comment 15: Another point to mention/develop would be the non-natural barriers (dams...).

Response: This is a good point. Our study region is void of any major dams that restrict salmon migration, as shown in the paper titled “High-resolution mapping of the world’s reservoirs and dams for sustainable river-flow management” by Lehner et al., 2011. We have included a sentence within the Methods section (lines 313-314) when introducing the present-day stream network highlighting the lack of major dams within our study region.

Reviewer 2, comment 16: I also believe it is crucial to develop the discussion on the importance of flow, temperature and glacial influence on salmon as well as the impacts of glacier retreat on these factors although those parameters could not be included in your models.

Response: Yes, there are multiple pathways that glacier retreat will impact river systems for salmon. We previously touched on this topic but have expanded more thoroughly within the main text including the different environmental factors that will impact the suitability of habitat for salmon. We have added a paragraph in the main text to address this topic (lines 219-230)

pdf specific comments.

Reviewer 2, comment L54: Among how many kms in total?

Response: *We do not explicitly comment on the total number of existing stream kms in this section of the paper, but in the following sentence we do state: “These increases in accessible stream kilometers represent 0 to 27% gains within the 18 sub-regions we studied”. This metric incorporates the size of the present-day river network (i.e., total kms), as it is a percent gain based on the present-day network size. Additionally, we included Extended Data Table 1 that includes metrics such as each sub-regions present-day river network size (kms).*

Reviewer 2, comment L54: hotspots changed to “habitat”

Response: *Change made.*

Reviewer 2, comment L70: Leading to how many kms of newly available streams?

Response: *We have incorporated the newly created river length into this sentence on lines 71-73. The sentence now reads “For example, pink salmon abundance grew to more than 5,000 adult spawners within ~15 years of a new stream (~2km) and lakes system being created following glacier retreat in Glacier Bay, Alaska (Milner et al., 2011)”.*

Reviewer 2, comment L108: rather 'up to a'

Response: *Corrected.*

Reviewer 2, comment L114: crossed out ‘that were larger in size’

Response: *Change made.*

Reviewer 2, comment L132: Could you rather provide values of glacier area?

Response: *We followed the great suggestion (Reviewer 2, comment 12) of including a table (Extended Data Table 1) with number of accessible glaciers and glacier area values for two stream gradient thresholds (10% and 15%) for each sub-region and the entire study region and have cited the table at the end of the suggested sentence.*

Reviewer 2, comment L134: To evaluate these values, we need to know the present-day stream network length.

Response: *Agreed. We have included the values of the present-day stream network length in Extended Data Table 1. We also reference the relative change in watershed river size throughout the paper, which incorporates the present-day stream network in that value and is communicated as a % change rather than an absolute value.*

Reviewer 2, comment L140: Also, the reason why I do not understand your 100m threshold.

Response: *We have removed the 100m threshold and have explained my reasoning in Reviewer 2, comment 4*

Reviewer 2, comment L143: Please, provide values to be able to evaluate this 'substantial' or 'relatively small' changes.

Response: We have provided Extended Data Tables 1 and 3 to help evaluate the changes for the total and each sub-region. Then we do elaborate on this point specifically, by providing an example using the Copper River. Given the large geographic region we considered, we found it helpful to use examples to explain some of the key findings.

Reviewer 2, comment L156: Where can we see these results?

Response: These results were not communicated in the previous version of the manuscript. We have addressed this shortcoming by creating Extended Data Table 3, which includes future stream kilometers for both 10% and 15% stream gradient thresholds using both the RCP 4.5 and 8.5 climate scenarios for each sub-region.

Reviewer 2, comment L169: This result is quite obvious and comes directly from your definition of the two different habitats. Not sure the information bring much here rather provide the results for the entire study zone.

Response: We have revised the text (lines 180-183) and added Extended Data Table 3 that contains the kilometer gain values.

Reviewer 2, comment L177: Sure, but probably not continuously, 500 to 1500 salmon produced per km, but probably not continuously along a stream. What is the frequency of those high-density stream reaches along a stream? Thereby, I do not believe the estimation of hundreds of thousands to millions salmon makes any sense.

Response: We appreciate this point, but we respectfully disagree. While there are undoubtedly areas that are more or less productive, the estimate of 500 to 1500 salmon produced per km was indeed from studies that examined salmon production across entire systems. Thus, the estimates we provide are based on aligned scales of study. Specifically, the findings of 500-1,500 smolts per km from Bradford et al 1997 were determined at the river scale from 83 different rivers over a large geographic area (see Figure below). However, we do highlight that there are many uncertainties and unknowns in predicting exact smolt abundance numbers from our results (future stream kilometers), therefore we refer to the potential gains in salmon abundance in these future rivers that will be created. Thus, why we communicate this finding as a potential within the discussion rather than as a result.

FIGURE 1.—Relation of mean coho salmon smolt abundance (Y) to stream length (X) for 83 streams and rivers (double \log_e plot).

Reviewer 2, comment L180: Already explained several times.

Response: Agreed. We have removed this sentence.

Reviewer 2, comment L183: Even though glaciers are not mentioned in this study.

Response: The rivers selected for this study were not explicitly glacier-fed. The database was assembled from 86 streams in western North America, which will likely contain some rivers that are indeed glacier-fed. The fact that this paper does not include glaciers as a covariate should not impact the relationship between stream length and smolt yield, which is the relationship we are highlighting in this section.

Reviewer 2, comment L184: Yes, this is your assumptions on which you have based your study, but you did not prove it in this paper.

Response: There is a strong existing body of research that has found that salmon do colonize rivers unveiled by glacier retreat (e.g., Milner et al., 2011), and a strong positive and linear relationship between stream length and salmon abundance (Bradford et al., 1997). Thus, we do believe that we have proven that with the increase in salmon habitat, there will likely be an increase in the number of salmon within these streams. To address this concern, we have expanded on caveats that may impact our findings as well within the main text (lines 194-198) and Methods section (lines 544-559).

Reviewer 2, comment L194: where does this value comes from?

Response: This is the percent of glaciers that are not accessible to salmon. 50% of the glacier area within our study region is accessible to salmon, and the remaining 50% is steep and beyond the accessibility of salmon. We have included these values in Extended Data Table 1 and reference this table at the end of this sentence.

Reviewer 2, comment L202: Do you know, for those glaciers, where the coming years are located along the meltwater flow evolution under glacier retreat? Are we before or after the threshold of reduction in glacial meltwater contribution to stream flow with glacier retreat?

Response: We think the reviewer is referring to the findings from Huss and Hock (2018) who project future glacier runoff changes for major watersheds around the world. The findings from that paper indicate that the Copper, Alsek, and Susitna watersheds (all in southcentral region of Alaska) show an increase in glacier meltwater during the summer months throughout the entire 21st century. Thus, we feel that in the sentence “southcentral Alaska regions, such as the Copper River, will experience substantial increases in glacier meltwater during the summer months over the coming decades due to the extensive network of glaciers that contribute flow to pro-glacial rivers (Huss and Hock, 2018)” explains these important findings. We also cite the Huss and Hock, 2018.

Reviewer 2, comment L204: Rather cite the diverse mechanisms.

Response: *We have elaborated on the impacts (i.e., diverse mechanisms) glacier retreat will have on downstream habitats by including a new paragraph. This paragraph is attached to Reviewer 2, comment 16.*

Reviewer 2, comment L284: Out of topic in this paragraph.

Response: *We have deleted this sentence.*

Reviewer 2, comment L305: Could you provide more information on this catalogue? Did you consider the upper point or the mouth of the water body for the presence of fish?

Response: *We had additional information in our previous version at the end of the paragraph, as mentioned in Reviewer2, comment L316, and we agree that the details of the database need to follow the initial introduction. Therefore, we have moved the description of the data source immediately following its introduction. For presence of fish, we used the spatial reference of the observed individual salmon species. We have clarified this point in the methods section on lines 339-341).*

Reviewer 2, comment L306: Might be good to perform this exercise in various watersheds to assess the variability across catchments... the effects of glacier influence to understand the bias.

Response: *This is a very interesting suggestion. We selected the Susitna River watershed to be used to validate the 10% and 15% migration thresholds given that it contains good coverage of salmon presence data and diversity in catchment types such as rain fed, snow fed, and glacier fed containing different salmon habitat types. The Susitna River watershed is complex, 53,366km², containing 1,278 glaciers, and a stream network of 34,336 km. We also selected the Susitna River watershed given its good coverage of salmon presence data, there is a minimum of ~200 individual salmon observations for each species. Other watersheds have poorer coverage of salmon data given the limited access to some of these watersheds to observe salmon. Additionally, the database only has coverage for Alaska and not British Columbia. Thus, we believe that this watershed provides a strong validation model to be applied to the other watersheds given its diversity and good salmon coverage. We have provided more information about this sample watershed within the methods section (lines 344-349), included the Richardson and Milner, 2005 reference citing the complexity of the watershed, and within Extended Data Table 1.*

Reviewer 2, comment L310: In the extended data Fig 1, you rather mentioned median stream gradient along stream segments from stream outlet to point data.

Response: *Thank you for catching this error. We have changed the Extended Data Fig 1 caption (lines 794-797) and provided a clearer explanation of the validation analysis used as explained in Reviewer 2, Comment 5.*

Reviewer 2, comment L313: Only 1 and 2 reaches > 15%, observed for chum and pink salmon, respectively => not the majority.

Response: *This is correct, it was supposed to read >10%. We have made the correction on lines 354-357).*

Reviewer 2, comment L314: Justification unclear to me.

Response: Agreed that this sentence was poorly communicated, we have revised the text on lines 357-359.

Reviewer 2, comment L316: To mention above while presenting the database.

Response: Very good point! I have moved the details of the AWC database following the first introduction of the database, as suggested in Reviewer 2, comment L305.

Reviewer 2, comment L317: What about Canada?

Response: There are no comparable regionally wide salmon presence (or absence) data available in Canada. The AWC only contains salmon presence data for the state of Alaska. Supporting why we ran or stream gradient validation on the Susitna River watershed.

Reviewer 2, comment L323: Why this threshold?

Response: We have provided a response in Reviewer2, comment 4

Reviewer 2, comment L330: Same DEMs as the ones used for present-day streams? If yes, please clarify, using similar term for example.

Response: No, this is a different DEM as the ones used to build the present-day stream network. Throughout the manuscript we have tried to keep the different dataset terminologies clear. For example, the DEM used for building the synthetic present-day river network are defined as the “ASTER global DEMs”, and the DEMs used for building the future river network are defined as the “ice surface DEMs”.

Reviewer 2, comment L331: Please indicated, in parenthesis, you will develop this point below

Response: corrected, as suggested

Reviewer 2, comment L333a: What do you mean?

Response: We have clarified that the dataset uses the glacier outlines from RGI v6.0 on lines 373-375.

Reviewer 2, comment L333b: How did you check it? The order of magnitude? Could you please clarify.

Response: We ran a direct comparison of local ice thicknesses and overall volumes between the data set used here (Huss and Farinotti, 2012, updated to RGIv6.0, model 1 contained in Farinotti et al., 2019) and the ice thickness consensus estimate (Farinotti et al., 2019). Our assessment indicates that the differences are small with respect to the other uncertainties that we account for. Total ice volume of glaciers bigger than 5 km² differs by less than 3% in the two relevant regions of the Randolph Glacier Inventory (Alaska, Western Canada, RGI01, RGI02). Also 90% to 99% quantile ice thicknesses, relevant for determining the bedrock slope of deglaciated valleys, only differ insignificantly (<10 m on average). An evaluation of local

thickness differences indicates a standard deviation relative to local thickness of about 18%. This value is likely influenced by outliers and still lies below the 25.9% uncertainty (Farinotti et al., 2019) that was very conservatively assumed to be systematic in our uncertainty analysis. We summarize this comparison in the Methods section on lines 375-378.

Reviewer 2, comment L335a: Validation on how many glaciers?

Response: Ice thickness was validated using point measurements from 300 glaciers from most glacierized regions of the world (lines 378-379).

Reviewer 2, comment L335b: Ice surface DEMs?

Response: Yes, corrected.

Reviewer 2, comment L335c: And same resolution (30m and 5m in vertical) for both DEMs? This DEMs was used for the present-day streams below 60°N? If yes, could you please mention it above?

Response: This DEM was not used for present-day streams, and rather just the ASTER DEM was used to build the present-day stream network. We have included in this section the spatial and vertical resolution of both DEMs (lines 379-383).

Reviewer 2, comment L342: I am not convinced of the use of stream order to estimate stream size; especially in your context where first-order stream below large glacier could be rather considerably wide.

Response: We understand the reviewer's concerns about the use of stream order and agree that the rivers that form downstream of the glacier snout will not be equivalent to a first order stream. However, given that we have modelled the river as a branching river network, the new river that will be created after disappearance of that glacier, in many cases, is indeed assigned > 2 order. To help clarify this point, in Figure 1 the black and blue lines – blue being segments >10% (migrating salmon), and blue being segments 0-2% (suitable habitat) – represent future salmon-accessible streams, are larger in the mainstem (stream order 2) with the tributaries being smaller in size (stream order 1). We have added this detail to the figure to help clarify the point better. Thus, while stream order is an imperfect proxy for stream size, given the data and spatial extent of our analyses we were constrained to this metric. First order stream are typically higher elevation rivers further up the deglaciated surface, and these streams will likely be less ideal for salmon. We believe that stream order provides, although coarse, a conservative estimate of rivers that will present habitat suitable for salmon.

Reviewer 2, comment L356: Might be clearer with you mention here (again) the need of DEMs and glacier outlines for the models.

Response: Corrected on lines 400-401

Reviewer 2, comment L386: where in the manuscript can we see the results for each GCM?

Response: *The results from the five individual GCMs are presented as the error bars in both Fig 2 and Extended Data Fig 2. In both figures, the captions state “Uncertainty corresponds to \pm one standard deviation and originate from: GCM projections (RCP4.5, for 2050 and 2100), ice thickness estimates, and stream segment length (see Methods section “Uncertainty estimates”)”. Given space constraints and requirements for clarity of the figures we are not able to present the results for all five GCMs individually. Furthermore, it is common to base impact studies on the mean of several GCMs to mitigate the uncertainties of individual models (e.g. Marzeion et al., 2020).*

Reviewer 2, comment L392: could you justify, also with references.

Response: *Yes, we have revised the text in the methods section (lines 434-437) and referenced Extended Data Table 2 that includes all applicable literature that supports these suggested thresholds.*

Reviewer 2, comment L403: I do not see what this extended fig brings in addition to fig 3.

Response: *We agree with this comment and have removed Extended Data Fig 3 from the manuscript and have referenced Fig 3.*

Reviewer 2, comment L414: sorry, I do not understand this argument.

Response: *It was not feasible due to limited computational resources to shorten our segment lengths beyond the ~500m segment length given the extent of the geographic region we were considering. In addition, using a 500 m segments minimizes errors/anomalies in the DEM one might observe using smaller stream segments. We have revised the text in the Methods section (lines 497-498) to better reflect this.*

Reviewer 2, comment L417: this sentence is also unclear to me.

Response: *We have deleted this sentence and have explained it in different terms in the previous sentences as per Reviewer 2, comment L414.*

Reviewer 2, comment L425: what about non-natural barriers?

Response: *Thank you for this comment, we have addressed this topic in Reviewer 2, Comment 15.*

Reviewer 2, comment L428: to evaluate the impact of segment length, I would directly examine the effect on the size of the stream network (as you mentioned below) instead of the number of accessible glaciers or both at the same time.

Response: *The same comment was made in Reviewer 2, comment 9. Please see our response above.*

Reviewer 2, comment L430: why only for 2 sub-regions? Especially as if below, you did it for all regions? And what are the characteristic of these two regions: hard to evaluate your analyses without the total area in km², and total stream length in km, the total number of glaciers...

Response: *We selected the two sample watersheds for assessing our validation analysis on the present-day river network. These datasets are large, and it was not feasible to run multiple iterations of this analysis due to limited computational resources. These two sample watersheds represent diversity in watershed characteristics such as terrain, relief, elevation, and represent other regions of our study area. To address the second question, we were able to run the validation analysis on future river kilometers on the entire study region as the computational power required was significantly less. We have added Extended Data Table 1 that has associated information suggested by the reviewer and included text to explain reasoning on lines 457-461.*

Reviewer 2, comment L435: rather to keep for discussion.

Response: *We have removed this sentence from this paragraph.*

Reviewer 2, comment L437: out of topic here.

Response: *We have removed this sentence from this paragraph.*

Reviewer 2, comment L447: rather because the measure would be more precise, a 750m <10% stream segment could include at least one 250m >10% stream segment.

Response: *The reviewer brings up a very good point. We have revised the text on lines 525-527.*

Reviewer 2, comment L449: see also my previous comment, the 750m segments could prevent the detection of small steep sections.

Response: *Yes, agreed. We have changed this sentence on lines 527-529, like suggested in Reviewer 2, comment L447.*

Reviewer 2, comment L451: merge both sentences.

Response: *Performed.*

Reviewer 2, comment L454: yes, even though 250m segment seems reasonable for a 30m DEMs.

Response: *Agreed that a 250 m segment seems reasonable for a 30m DEM. However, our analysis is much more nuanced than this one specific value. For the sensitivity analysis, which the reviewer is referring to, we were using the future stream network modelled from deglaciated terrain (ice thickness subtracted from ice surface DEMs). In some instances, the resolution ranged up to 200m, as stated on line 530. We also wanted to ensure that the present-day stream network and the future stream network stream gradients were derived from the same segment lengths for comparison purposes, and we ran into computational memory issues (within ArcGIS) when we reduced our segment length given the large geographic area we were considering. We have made sure that these limitations were clearly communicated throughout the manuscript (lines 529-533).*

Reviewer 2, comment L455: >10 % is not insignificant!

Response: *Agreed that 10% is not insignificant. We have removed this statement from the manuscript and have incorporated other sources of error into a global uncertainty as mentioned in my response to Reviewer 2, comment 1 and Reviewer 2, Comment L454.*

Reviewer 2, comment L457: what do you mean here? fine DEM resolution?

Response: *This was unclear phrasing. We have better communicated the data limitations, as per my Response to Reviewer 2, Comment L454.*

Reviewer 2, comment L465: same DEMs, right? And, If I am not wrong, this is the first time you mentioned this 200m resolution. Are you referring to the gridded ice thickness data? What about the altitude resolution in this case? This is an important issue that need to be developed and mentioned earlier.

Response: *Yes, these are indeed the ice surface DEMs. We mention the resolution previously on line 371 and have added another section about the ice surface DEMs in the “Uncertainty estimates” section on line 530.*

Reviewer 2, comment L475: you meant past?

Response: *Sorry for the confusion. Here we are referencing how the IPCC climate projection models used in this manuscript have proven to be good at predicting changes in climate quite accurately. We have removed the “future” and replaced it with “forecasted”.*

Reviewer 2, comment L478: no standard deviation here.

Response: *We referenced the incorrect figure number. Thank you for catching this. We have changed this to Figs 2 and 3.*

Reviewer 2, comment L486: ok, but there might still be errors with the first stream segment between the present-day streams and the future salmon accessible streams.

Response: *The analyses are conducted separately. Therefore, the present-day stream network ends (derived from surface DEMs), then the future stream network begins (derived from the ice thickness and ice surface DEMs). We suggest that the error in the estimated elevation of the bedrock is likely similar at either end of the ~500m segments, which remains correct as even at the very start of the future stream network (where present-day network butts against it), the segments are still derived from the ice thickness and ice surface DEMs. So, the error remains consistent across the glacier profile, therefore we have made no changes to this sentence.*

Reviewer 2, comment L715: 10% in the fig legend

Response: *Corrected, thank you.*

Reviewer #3 (Remarks to the Author):

This manuscript presents the results of a study linking climate change, glacial retreat, and increases in potential Pacific salmon habitat. Combining climate model-driven estimates of glacial loss and physical thresholds for salmon migration, spawning and rearing, the authors conclude that glacial retreat could open up a large amount of potential habitat with the potential to provide compensation for losses elsewhere with the current geographical range of Pacific salmon. While the idea of northern distributional expansion has been much discussed, and in some cases documented in the general ecological literature, this is one of the few studies to quantify the likely magnitude of the effect and link it to a specific climate-driven mechanism. Given this novelty, general potential implications, and the ecological, economic, and cultural importance of salmon, I think that the manuscript meets the standards and requirements of the journal.

I found it clear and easy to follow, and have only a few relatively minor comments:

Reviewer 3 comment 1: While some reference is made to the generality of potential northern expansion across species and systems, I still found the paper to be a bit salmon-centric for Nature and encourage the authors to think more broadly.

Response: *We have included a sentence in the introduction (lines 61-63) that broadens the effects of climate change and ice loss to other systems.*

Reviewer 3 comment 2: The authors make a very good point about the need to think pro-actively in terms of conservation and protection of these new habitats (which could again be a more general consideration across many species for northern expansion). I wonder if the results also suggest the particular importance of the adjacent source populations and their habitats in providing colonists.

Response: *This is an excellent observation by the reviewer. We have added a sentence to the main text (lines 223-225) to highlight this. We cover this topic in detail in the synthesis article (Pitman et al. 2020 BioScience).*

Reviewer 3 comment 3: I understand that the authors cannot include all the complex details of channel evolution following glacial retreat into their models and analyses. However, I was curious about the relationship between channel width (with its associated influence on total habitat area) and the habitats provided by glacial retreat. If new channels tend to be narrow, while existing habitat includes quite a lot of wide channels further down in the drainage network, how will this affect the relative importance of new habitat (essentially will the proportional contribution of new habitat be lessened if area not length is the dimension considered)?

Response: *This is an excellent question. However, and perhaps surprisingly, there is not a strong relationship between salmon habitat area and channel width in moderate to large sized streams. This is because most usable habitat area in these streams tends to occur on the edges of the stream channel. High thalweg velocities often limit juvenile rearing to habitat patches at the edge of streams (see Murphy et al. 1989 for an excellent example in the glacial Taku River). Similar patterns are observed for adult spawning (Lorenz and Eiler 1989). Consequently, coarse-scale estimates of juvenile production potential are not very sensitive to systematic*

increases in stream width. The use of linear extent of stream habitat to estimate coarse-scale smolt and adult salmon abundances has strong empirical support (see Bradford et al. 1997, Liermann et al. 2010).

Reviewers' Comments:

Reviewer #1:

Remarks to the Author:

The authors have responded to my comments/suggestions thoughtfully and thoroughly, and have made changes to the manuscript to reflect this where appropriate. I recommend the manuscript to be accepted for publication.

Reviewer #4:

Remarks to the Author:

This is a very interesting manuscript.

One of the main concern of reviewers (1 and 2) was the uncertainty assessment. Authors reviewed and commented all the questions of the previews reviewers. It seems that all questions have been sufficiently clarified, especially uncertainty.

My objective was mainly to review the responses of the authors to the comments (n=16 main comments and lots of specific comments) of Reviewer #2 dealing with habitat topic.

Habitat is roughly described by slope and river length (containing successions of morphological units such as riffle, pool...). One threshold was defined for each variable (one value of slope for colonization and two values of slope for suitable habitat of both salmon live stages), from available data and/or literature. Those thresholds were discussed with a sensitivity analysis. Of course fish habitat is more complex than these both variables, as underlined by the authors. But temporal and spatial scales do not enable the authors to be more accurate about habitat. If discharge could be simulated for different period of the year for the "new accessible stream kilometers" available and suitable new hydraulic habitat could be more precisely predicted.

Authors proposed a clear response to the Reviewer #2 – comment 4, 5, 6, 9, 11, 14 and 16. See also response to Reviewer 2, comment L177 (results were only "potential" abundance; this potential abundance is useful to compare different sub-regions).

Although thresholds were tested with sensitivity analysis no proof of the relationship (cause and effect) between such habitat features (slope and length) and salmon presence / abundance was proposed by authors. The Figure of Bradford et al. 1997 (see response of Reviewer 2, comment L177) came from 83 different rivers, but certainly none of them were newly colonized rivers (e.g. after glacier retreat). Thresholds have to be tested as soon as possible (e.g. in newly available habitats due to glacier retreat) to be validated or modified. I think this is an essential step before the future bargaining process with gold mine owners.

In my opinion, authors adequately addressed comments from reviewers, specifically those from Reviewer #2 dealing with habitat.

REVIEWERS' COMMENTS

Reviewer #1 (Remarks to the Author):

Reviewer 1, comment 1: The authors have responded to my comments/suggestions thoughtfully and thoroughly and have made changes to the manuscript to reflect this where appropriate. I recommend the manuscript to be accepted for publication.

Response: Thank you!

Reviewer #4 (Remarks to the Author):

Reviewer 4, comment 1: This is a very interesting manuscript. One of the main concern of reviewers (1 and 2) was the uncertainty assessment. Authors reviewed and commented all the questions of the previews reviewers. It seems that all questions have been sufficiently clarified, especially uncertainty. My objective was mainly to review the responses of the authors to the comments (n=16 main comments and lots of specific comments) of Reviewer #2 dealing with habitat topic.

Response: Thank you!

Reviewer 4, comment 2: Habitat is roughly described by slope and river length (containing successions of morphological units such as riffle, pool...). One threshold was defined for each variable (one value of slope for colonization and two values of slope for suitable habitat of both salmon live stages), from available data and/or literature. Those thresholds were discussed with a sensitivity analysis. Of course fish habitat is more complex than these both variables, as underlined by the authors. But temporal and spatial scales do not enable the authors to be more accurate about habitat. If discharge could be simulated for different period of the year for the “new accessible stream kilometers” available and suitable new hydraulic habitat could be more precisely predicted.

Response: We agree that fish habitat is complex beyond the variables we used in our paper, given the scale of the paper. Data, such as discharge, is not possibly obtained for the future rivers that are currently under ice. However, we do refer to this limitation of our analysis in the Results and Discussion section (lines 190-193) and elaborate in more detail in the Uncertainties section under the heading “Unknown potential changes in habitat and landscape variables important to salmon spawning and rearing” on lines 510-525.

Reviewer 4, comment 3: Authors proposed a clear response to the Reviewer #2 – comment 4, 5, 6, 9, 11, 14 and 16. See also response to Reviewer 2, comment L177 (results were only “potential” abundance; this potential abundance is useful to compare different sub-regions). Although thresholds were tested with sensitivity analysis no proof of the relationship (cause and effect) between such habitat features (slope and length) and salmon presence / abundance was proposed by authors. The figure of Bradford et al. 1997 (see response of Reviewer 2, comment

L177) came from 83 different rivers, but certainly none of them were newly colonized rivers (e.g. after glacier retreat). Thresholds have to be tested as soon as possible (e.g. in newly available habitats due to glacier retreat) to be validated or modified. I think this is an essential step before the future bargaining process with gold mine owners.

***Response:** Yes, the 83 rivers that were sampled in the Bradford et al. 1997 paper were presumably not from rivers immediately following glacier retreat. Given the range of latitudes of systems, it is highly probable that these rivers include those that are glaciated. Bradford et al. 1997 surprisingly does not actually list the specific study rivers, which precludes further investigation of this. Salmon abundance data from newly exposed streams are not available - the only reported salmon abundance estimates were from the work of Milner et al, 2011, which we cite throughout the paper. Additionally, in this paragraph we are not only assuming salmon population growth following immediate retreat of glaciers, and rather that these rivers will become future hot spots for salmon habitat and productivity over times as the glaciers to recede from the landscape.*

Reviewer 4, comment 4: In my opinion, authors adequately addressed comments from reviewers, specifically those from Reviewer #2 dealing with habitat.

***Response:** Thank you!*